# WebThinker: Empowering Large Reasoning Models with Deep Research Capability

**Xiaoxi Li**[1*], **Jiajie Jin**[1*], **Guanting Dong**[1*], **Hongjin Qian**[2], **Yongkang Wu**[3]
**Ji-Rong Wen**[1], **Yutao Zhu**[1†], **Zhicheng Dou**[1†]
[1]Renmin University of China   [2]BAAI   [3]Huawei Poisson Lab
{xiaoxi_li, ytzhu, dou}@ruc.edu.cn

## Abstract

Large reasoning models (LRMs), such as OpenAI-o1 and DeepSeek-R1, demonstrate impressive long-horizon reasoning capabilities. However, their reliance on static internal knowledge limits their performance on complex, knowledge-intensive tasks and hinders their ability to produce comprehensive research reports requiring synthesis of diverse web information. To address this, we propose WebThinker, a deep research agent that empowers LRMs to autonomously search the web, navigate among web pages, and draft reports during the reasoning process. WebThinker integrates a Deep Web Explorer module, enabling LRMs to dynamically search, navigate, and extract information from the web when encountering knowledge gaps. It also employs an Autonomous Think-Search-and-Draft strategy, allowing the model to seamlessly interleave reasoning, information gathering, and report writing in real time. To further enhance research tool utilization, we introduce an RL-based training strategy via iterative online Direct Preference Optimization (DPO). Extensive experiments on complex reasoning benchmarks (GPQA, GAIA, WebWalkerQA, HLE) and scientific report generation tasks (Glaive) demonstrate that WebThinker significantly outperforms existing methods and strong proprietary systems. Our approach enhances LRM reliability and applicability in complex scenarios, paving the way for more capable and versatile deep research systems. The code is available at https://github.com/RUC-NLPIR/WebThinker.

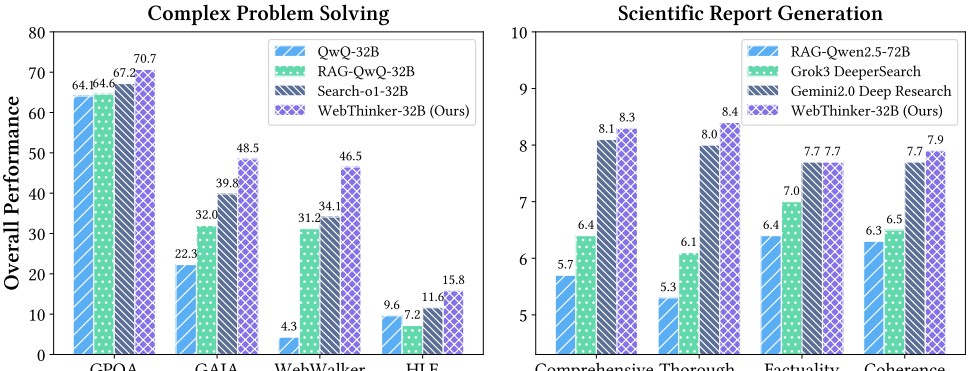

Figure 1: Overall performance comparison of WebThinker with other models across two tasks: complex reasoning problem-solving (left) and scientific research report generation (right).

*Equal contribution.
†Corresponding authors.

39th Conference on Neural Information Processing Systems (NeurIPS 2025).

# 1 Introduction

Recently, large reasoning models (LRMs) have demonstrated remarkable capabilities across various domains like math, code, and science [34, 48, 14]. However, when confronted with complex information research needs, models that rely solely on internal knowledge struggle to conduct in-depth web information retrieval and to generate comprehensive and accurate reports through multi-step reasoning. Therefore, the deep integration of LRMs' reasoning capabilities with web information exploration has become a practical demand, which has sparked a series of deep research initiatives by OpenAI [35], xAI Grok3 [12], and Google Gemini [10].

The objective of deep research technology is quite revolutionary: enabling users to conduct deep searches, mining, and integration of comprehensive and reliable research information across the internet's massive information landscape through simple queries. This approach significantly reduces the time and costs associated with information gathering for researchers in knowledge-intensive fields (e.g., finance, science, engineering).

Unfortunately, existing open-source deep search agents typically employ retrieval-augmented generation (RAG) techniques with predefined workflows [1, 63, 42], which limits LRMs' ability to explore deeper web information and hinders close interaction between LRMs and search engines (Figure 2 (a) & (b)). Consequently, developing a universal, flexible, open-source deep research framework has emerged as a critical challenge urgently awaiting resolution in both academic and industrial circles.

To address this, we propose **WebThinker**, an autonomous deep research agent entirely powered by large reasoning models, as illustrated in Figure 2 (c). It enables LRMs to autonomously conduct web searches and web page navigation to acquire external knowledge during the reasoning process, facilitating complex real-world problem solving. Furthermore, WebThinker allows LRMs to draft reports concurrently with thinking and searching. Once sufficient information is gathered for a specific section, the model can draft that part, ultimately producing comprehensive and customized reports tailored to users' research questions.

To empower LRMs with the capability to deeply explore web information, we design a **Deep Web Explorer** module that enables LRMs to search, navigate pages by clicking interactive elements (like links or buttons), and extract relevant information. Based on the current query's search results, the LRM can initiate follow-up searches and traverse deeper links until it collects all relevant information.

To facilitate scientific report writing, we introduce an **Autonomous Think-Search-and-Draft** strategy that deeply integrates report writing with the reasoning and search processes of LRMs. Rather than generating the entire report all at once after searching, our approach enables the model to draft and seek necessary knowledge in real-time. To achieve this, we equip LRMs with three specialized tools: (1) drafting content for specific chapters, (2) checking the current report, and (3) editing the report. These tools enable LRMs to autonomously enhance the report by maintaining comprehensiveness, coherence, and adaptability to newly discovered information during reasoning.

To further unlock the deep research potential of LRM backbones, we develop **RL-based training strategies** to optimize end-to-end task performance. We leverage the LRM equipped with the research tools to sample large-scale reasoning trajectories from complex tasks [46, 56, 47, 65, 25, 11]. Based on the accuracy of reasoning, tool usage, and final outputs, we construct preference pairs for online DPO training [40, 57, 5]. Through iterative, on-policy RL training, the model progressively improves its ability to perceive, reason, and interact with research tools effectively.

We conduct extensive experiments on (1) knowledge-intensive complex reasoning benchmarks, including GPQA [41], GAIA [32], WebWalkerQA [56], and Humanity's Last Exam (HLE) [37] to assess complex problem-solving capabilities, and (2) open-ended reasoning tasks from Glaive [11] to evaluate report quality. As Figure 1 shows, WebThinker consistently outperforms all competing approaches. Specifically, WebThinker surpasses Search-o1 by 21.9% and 36.2% on the GAIA and HLE datasets, respectively, and even outperforms Grok3 and Gemini2.0 on report generation tasks.

# 2 Related Work

**Large Reasoning Models.** Large reasoning models (LRMs) enhance test-time performance through extended reasoning, setting them apart from traditional large pre-trained models that scale mainly with model size or data volume [15, 61, 62, 28, 3]. Models like OpenAI-o1 [20], Qwen-QwQ [48], and

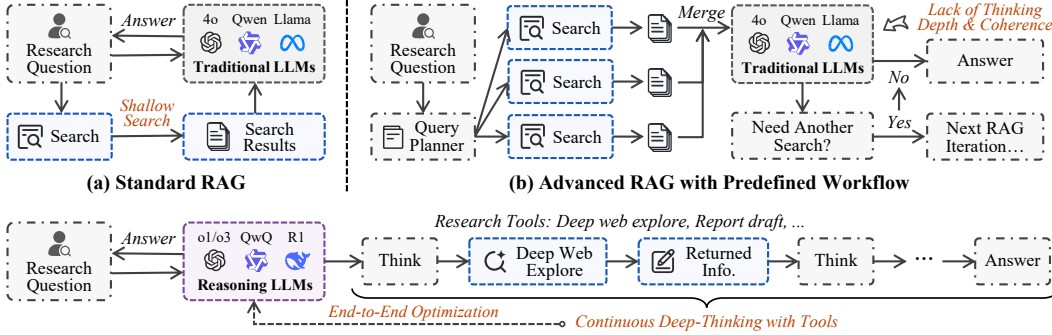

Figure 2: Comparison of RAG paradigms: (a) Standard RAG workflow, (b) Iterative RAG workflow, and (c) **WebThinker**, a reasoning agent that **autonomously searches, deeply explores web pages, and drafts research reports**, all within its continuous thinking process.

DeepSeek-R1 [14] demonstrate explicit chain-of-thought (CoT) reasoning [55], resembling human system-2 thinking in tasks such as math and programming [28]. Various strategies have been proposed to achieve o1-like reasoning capabilities [9, 4, 45, 59, 16]. Some approaches introduce intentional reasoning errors during training to help models partially internalize reasoning patterns [39, 64]. Others enhance reasoning ability through distilled training data [33]. More recently, reinforcement learning has been explored as a means to develop long CoT abilities in LLMs [14, 59, 16]. However, these methods are constrained by their reliance on static, parameterized architectures that lack access to external world knowledge. This limitation becomes particularly problematic in complex reasoning tasks requiring extensive real-world information.

**Retrieval-Augmented Generation.**   Retrieval-augmented generation (RAG) enhances generative models by integrating retrieval mechanisms, enabling access to external knowledge beyond static parameters [24, 66]. Recent advances cover multiple dimensions, including retrieval necessity [44], query reformulation [31, 52, 69], document compression [60, 23, 68], denoising [30, 21], and instruction-following [6]. Moreover, complex workflows such as structured planning [51, 58, 17] and decision-making frameworks [29, 13] have shown notable gains in multi-hop reasoning, planning, and domain-specific tasks [53]. Recent work has also explored integrating o1-style reasoning with retrieval. For example, Search-o1 [26] incorporates an agentic RAG framework and a Reason-in-Documents module to merge retrieval with reasoning via prompt engineering. Other studies employ reinforcement learning to train reasoning with search capabilities from scratch [22, 43, 2, 67, 50, 8, 27, 38, 54], showing strong results in Wikipedia-based QA. Nonetheless, these methods fall short in adapting to complex real-world reasoning scenarios and comprehensive report-writing tasks.

## 3   Methodology

### 3.1   Problem Formulation

We consider a complex reasoning task that requires both multi-step reasoning and the utilization of research tools. The objective is to generate a comprehensive answer solution for a given task query $q$, guided by an instruction $I$. A solution comprises a logical reasoning chain $\mathcal{R}$ and a final output $y$ (which could be an answer or a signal indicating completion). WebThinker enables the reasoning model to autonomously invoke tools from an available set $\mathcal{T}$ during its reasoning process, which can be formalized as the mapping $(I, q, \mathcal{T}) \rightarrow (\mathcal{R}, y)$. The generation process can be expressed as:

$$P(\mathcal{R}, y \mid I, q, \mathcal{T}) = \underbrace{\prod_{t=1}^{T_r} P(\mathcal{R}_t \mid \mathcal{R}_{<t}, I, q, \{\mathcal{O}_\tau\}_{\tau<t})}_{\text{Reasoning with Tools}} \cdot \underbrace{\prod_{t=1}^{T_y} P(y_t \mid y_{<t}, \mathcal{R}, I, q)}_{\text{Final Output Generation}}, \quad (1)$$

where $T_r$ is the number of tokens in the reasoning sequence $\mathcal{R}$. The token at position $t$ is $\mathcal{R}_t$, and $\mathcal{R}_{<t}$ represents all tokens generated before position $t$. $\{\mathcal{O}_\tau\}_{\tau<t}$ denotes the outputs of all tool calls made before position $t$. Similarly, $T_y$ is the length of the output sequence $y$, with $y_t$ being the token at position $t$ and $y_{<t}$ indicating all generated output tokens before position $t$.

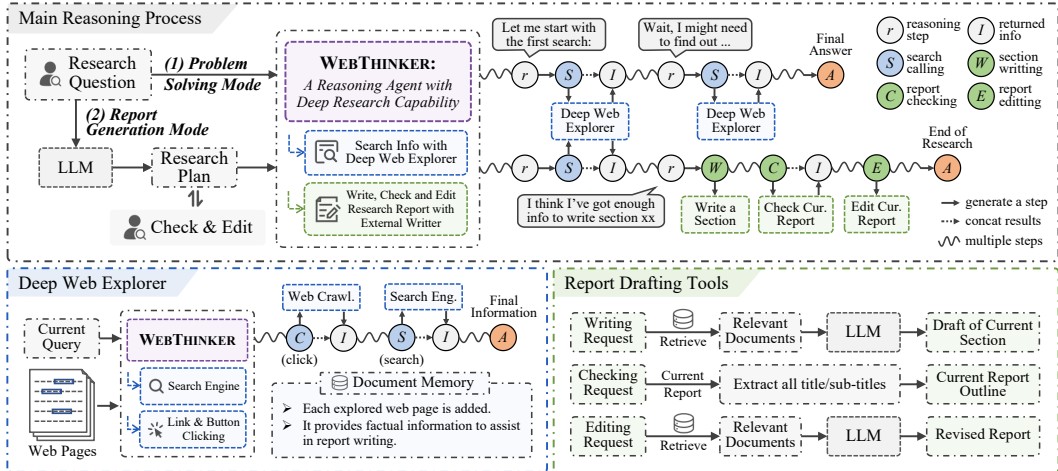

Figure 3: Overview of the WebThinker framework. It operates in two modes: **(1) Problem-Solving Mode** equips reasoning models with a search tool backed by a Deep Web Explorer, enabling thorough web exploration to retrieve relevant information for solving complex real-world problems. **(2) Report Generation Mode** extends the model's capabilities with writing, checking, and editing capabilities, allowing it to iteratively craft comprehensive research reports while thinking and searching.

## 3.2 Overview of the WebThinker Framework

WebThinker is designed to enhance large reasoning models with deep research capabilities by enabling autonomous web exploration and report generation during the reasoning process. As illustrated in Figure 3, WebThinker operates in two primary modes:

- **Problem-Solving Mode:** Empowers the LRM with a **Deep Web Explorer** module. When encountering knowledge gaps, the LRM can autonomously initiate web searches, navigate through web pages by clicking links or buttons, and extract relevant information before continuing its reasoning. This facilitates in-depth information gathering beyond standard shallow search.
- **Report Generation Mode:** Implements an **Autonomous Think-Search-and-Draft** strategy. The LRM interleaves reasoning, information seeking (via the Deep Web Explorer), and report generation. It utilizes specialized tools for drafting, checking, and editing report sections, executed by an assistant LLM leveraging an explored document memory, to ensure the final report is comprehensive, coherent, and grounded in the gathered evidence.

## 3.3 Solving Complex Reasoning Tasks with the Deep Web Explorer

In the Problem-Solving Mode, WebThinker tackles complex tasks requiring knowledge beyond the LRM's internal parameters. The core component is the Deep Web Explorer tool, $\mathcal{T}_{\exp} \in \mathcal{T}$, which the LRM can invoke during reasoning (Figure 3).

Given a task-specific instruction $I_q$ and query $q$, the LRM generates a reasoning chain $\mathcal{R}$, potentially interspersed with calls to the Deep Web Explorer. The final output in this mode is typically a direct answer $a$. The generation process is formalized as:

$$P(\mathcal{R}, a \mid I_q, q) = \prod_{t=1}^{T_r} P(\mathcal{R}_t \mid \mathcal{R}_{<t}, I_q, q, \{\mathcal{O}_{\exp}^{(j)}\}_{j<i(t)}) \cdot \prod_{t=1}^{T_a} P(a_t \mid a_{<t}, \mathcal{R}, I_q, q), \quad (2)$$

where $a = (a_1, \ldots, a_{T_a})$ is the answer sequence. $\{\mathcal{O}_{\exp}^{(j)}\}_{j<i(t)}$ denotes the set of outputs from all Deep Web Explorer calls completed before reasoning step $t$.

The Deep Web Explorer itself is driven by the LRM, operating under a specific instruction $I_e$. It utilizes two elementary tools: a search engine $\mathcal{T}_s$ to retrieve web pages based on a generated query $q_s$, and a navigation tool $\mathcal{T}_n$ to interact with elements (e.g., click links) on the currently viewed page(s) $\mathcal{D}$. The explorer, triggered by an information need $q_s$, generates its own internal reasoning chain $\mathcal{R}_e$ to decide whether to search further or navigate deeper based on the evolving web content $\mathcal{D}_t$ it encounters. Its goal is to produce a concise output $\mathcal{O}_{\exp}$ that addresses the knowledge gap in the main

reasoning chain $\mathcal{R}$. The exploration process within the tool is modeled as:

$$P(\mathcal{R}_{\text{e}}, \mathcal{O}_{\text{exp}} \mid q_{\text{s}}, \mathcal{D}, I_e) = \prod_{t=1}^{T_e} P(\mathcal{R}_{\text{e},t} \mid \mathcal{R}_{\text{e},<t}, q_{\text{s}}, \mathcal{D}_t, I_e) \cdot P(\mathcal{O}_{\text{exp}} \mid \mathcal{R}_{\text{e}}, q_{\text{s}}, \mathcal{D}, I_e), \quad (3)$$

where $T_e$ is the length of the explorer's reasoning chain $\mathcal{R}_{\text{e}}$. $\mathcal{D}_t$ represents the web content available at step $t$, which dynamically changes based on search and navigation actions. This hierarchical structure allows the main reasoning process to delegate complex information gathering tasks to the Deep Web Explorer, which can recursively search and navigate the web.

### 3.4 Generating Comprehensive Reports via Autonomous Think-Search-and-Draft

In the Report Generation Mode, the LRM autonomously produces comprehensive reports by interleaving reasoning, searching, and writing. Besides the Deep Web Explorer ($\mathcal{T}_{\text{exp}}$) for knowledge acquisition, the LRM utilizes a set of report writing tools $\mathcal{T}_{\text{write}} = \{\mathcal{T}_{\text{draft}}, \mathcal{T}_{\text{check}}, \mathcal{T}_{\text{edit}}\}$. These tools are implemented by an assistant LLM, separating the complex task orchestration performed by the main LRM from the detailed text manipulation required for report writing.

All web pages explored via $\mathcal{T}_{\text{exp}}$ are accumulated in a document memory $\mathcal{M}$. When the main LRM decides to invoke a writing tool (e.g., $\mathcal{T}_{\text{draft}}$), it generates an editing instruction $e$. The assistant model then receives $e$, the current report state $r$, and relevant documents $\mathcal{D}_{\text{top-k}}$ retrieved from $\mathcal{M}$. It produces the updated report content $r_{\text{new}}$ according to:

$$P(r_{\text{new}} \mid e, \mathcal{D}_{\text{top-k}}, r) = \prod_{t=1}^{T_{r_{\text{new}}}} P(r_{\text{new},t} \mid r_{\text{new},<t}, e, \mathcal{D}_{\text{top-k}}, r), \quad (4)$$

where $T_{r_{\text{new}}}$ is the length of the newly generated/edited report content $r_{\text{new}}$.

The main LRM's role is to orchestrate the overall process: performing reasoning steps, deciding when to explore for more information using $\mathcal{T}_{\text{exp}}$, and determining when and how to modify the report using $\mathcal{T}_{\text{write}}$. The main reasoning process concludes when the LRM generates the EOS token, denoted as $y_{\text{end}}$. This overall process, conditioned on the initial instruction $I$ and query $q$, is formalized as:

$$P(\mathcal{R}, y_{\text{end}} \mid I, q) = \prod_{t=1}^{T_r} P(\mathcal{R}_t \mid \mathcal{R}_{<t}, I, q, \{\mathcal{O}_{\text{exp}}^{(j)}\}_{j<i(t)}) \cdot P(y_{\text{end}} \mid \mathcal{R}, \mathcal{M}), \quad (5)$$

where $\{\mathcal{O}_{\text{exp}}^{(j)}\}_{j<i(t)}$ represents outputs from prior Deep Web Explorer calls. The document memory $\mathcal{M}$ serves as the knowledge base for the assistant LLM (Eq. 4) executing the writing operations.

### 3.5 Improving LRMs with Research Tools via Reinforcement Learning

To enhance the LRM's ability to effectively utilize research tools that range from high-level web exploration and report manipulation to elementary search and navigation actions, we employ on-policy RL training. This focuses on constructing preference data reflecting desired tool usage patterns and applying an iterative online DPO [40, 57, 5] strategy.

**Preference Data Construction.** We generate diverse reasoning trajectories using WebThinker on complex datasets (e.g., SuperGPQA [46], WebWalkerQA [56], OpenThoughts [47], NaturalReasoning [65], NuminaMath [25], and Glaive [11]). For each query $q$, the LRM self-samples $n$ distinct trajectories $\{\mathcal{R}^{(i)}\}_{i=1}^n$, capturing varied solution strategies and tool usage patterns across the main reasoning chain and the Deep Web Explorer's internal operations.

Our goal is to identify trajectories that demonstrate not only correctness but also efficient use of the research tools. To achieve this, we establish preference pairs $(\mathcal{R}_w, \mathcal{R}_l)$, where $\mathcal{R}_w$ is the preferred trajectory and $\mathcal{R}_l$ is the dis-preferred trajectory. We apply the following criteria iteratively in order of priority to pairs of sampled trajectories $(\mathcal{R}_i, \mathcal{R}_j)$ for the same task $q$:

1. **Overall Correctness/Report Quality:** If $\mathcal{R}_i$ yields a correct final answer (for reasoning tasks) or a higher quality final report (for report generation tasks), while $\mathcal{R}_j$ does not, then $\mathcal{R}_i$ is preferred ($\mathcal{R}_w = \mathcal{R}_i, \mathcal{R}_l = \mathcal{R}_j$). This rule takes precedence over all others.
2. **Tool Efficiency:** If both $\mathcal{R}_i$ and $\mathcal{R}_j$ reach the correct final answer, the trajectory with fewer total tool calls is preferred. If total_tool_calls($\mathcal{R}_i$) < total_tool_calls($\mathcal{R}_j$), then $\mathcal{R}_w = \mathcal{R}_i, \mathcal{R}_l = \mathcal{R}_j$.
3. **Thinking Conciseness:** If both $\mathcal{R}_i$ and $\mathcal{R}_j$ are correct and involve the same number of tool calls, the shorter trajectory is preferred when the length ratio exceeds the threshold $\gamma > 1$. If $\text{len}(\text{output}_j)/\text{len}(\text{output}_i) > \gamma$, then $\mathcal{R}_w = \mathcal{R}_i, \mathcal{R}_l = \mathcal{R}_j$.

Table 1: Main results on challenging research tasks, including PhD-level science QA, general AI assistants, and web exploring benchmarks. We report Pass@1 metric for all tasks. For 32B models, the best results are in **bold** and the second are underlined. Results from larger or closed-sourced models are in gray color for reference. '†' denotes results from their official releases.

| Method | GPQA (Science QA) | | | | GAIA (General AI Assist.) | | | | WebWalkerQA | | | |
|---|---|---|---|---|---|---|---|---|---|---|---|---|
| | Phy. | Chem. | Bio. | Avg. | Level 1 | Level 2 | Level 3 | Avg. | Easy | Med. | Hard | Avg. |
| *Direct Reasoning (w/o Retrieval)* | | | | | | | | | | | | |
| Qwen2.5-32B | 52.3 | 30.1 | 68.4 | 43.4 | 20.5 | 9.6 | 8.3 | 13.6 | 3.8 | 2.5 | 3.3 | 3.1 |
| DeepSeek-R1-32B | 82.5 | 41.9 | 73.7 | 62.6 | 23.1 | 17.3 | 0.0 | 17.5 | 7.5 | 1.4 | 4.2 | 3.8 |
| QwQ-32B | 84.8 | 44.1 | 68.4 | 64.1 | 30.8 | 15.4 | **25.0** | 22.3 | 7.5 | 2.1 | 4.6 | 4.3 |
| Qwen2.5-72B | 58.1 | 39.8 | 57.9 | 49.5 | 20.5 | 13.5 | 0.0 | 14.6 | 9.4 | 7.1 | 3.3 | 6.3 |
| GPT-4o | 62.8 | 46.2 | 68.4 | 55.6 | 23.1 | 15.4 | 8.3 | 17.5 | 6.7 | 6.0 | 4.2 | 5.5 |
| DeepSeek-R1-671B | 90.7 | 57.0 | 84.2 | 74.2 | 43.6 | 26.9 | 8.3 | 31.1 | 5.0 | 11.8 | 11.3 | 10.0 |
| o1-preview† | 89.4 | 59.9 | 65.9 | 73.3 | - | - | - | - | 11.9 | 10.4 | 7.9 | 9.9 |
| *Enhancing Reasoning with RAG Workflow* | | | | | | | | | | | | |
| RAG-Qwen2.5-32B | 59.3 | 41.9 | 68.4 | 52.0 | 12.8 | 11.8 | 8.3 | 11.8 | 23.1 | 14.3 | 11.3 | 15.3 |
| w/ Query Planning | 61.6 | 40.9 | 52.6 | 51.0 | 30.8 | 17.3 | 0.0 | 20.4 | 29.4 | 36.4 | 25.0 | 30.7 |
| w/ Iterative RAG | 64.0 | 41.9 | 57.9 | 53.0 | 35.9 | 19.2 | 8.3 | 24.3 | 30.6 | 35.7 | 25.4 | 30.9 |
| RAG-QwQ-32B | 84.9 | 46.2 | 63.2 | 64.6 | 33.3 | 36.5 | 8.3 | 32.0 | 36.9 | 26.1 | 33.5 | 31.2 |
| w/ Query Planning | 87.2 | 46.2 | 68.4 | 66.2 | 48.7 | 25.0 | 8.3 | 32.0 | 28.8 | 35.7 | 30.8 | 32.5 |
| w/ Iterative RAG | 84.9 | 45.2 | 73.7 | 65.2 | 51.3 | 28.8 | 8.3 | 35.0 | 29.4 | 32.9 | 31.3 | 31.5 |
| *Autonomous Search within Reasoning* | | | | | | | | | | | | |
| OpenAI Deep Research† | - | - | - | - | 74.3 | 69.1 | 47.6 | 67.4 | - | - | - | - |
| Search-o1-32B | 84.9 | 49.5 | 73.7 | 67.2 | 53.8 | 34.6 | 16.7 | 39.8 | 43.1 | 35.0 | 27.1 | 34.1 |
| WebThinker-32B-Base | 87.2 | **51.6** | 68.4 | 68.7 | 53.8 | 44.2 | 16.7 | 44.7 | 47.5 | 41.1 | 39.2 | 41.9 |
| WebThinker-32B-RL | **90.7** | 50.5 | **78.9** | **70.7** | **56.4** | **50.0** | 16.7 | **48.5** | **58.8** | **44.6** | **40.4** | **46.6** |

By applying these rules across all valid sampled trajectory pairs for all tasks, we construct a collection $\mathcal{D} = \{(I, q, \mathcal{R}_w, \mathcal{R}_l)_k\}$ of preference pairs.

**Iterative Online DPO Training.** We utilize the constructed preference dataset $\mathcal{D}$ to train the LRM using iterative online DPO. The standard DPO loss function aims to increase the likelihood of preferred trajectories $\mathcal{R}_w$ while decreasing the likelihood of dispreferred trajectories $\mathcal{R}_l$, relative to a reference policy $\pi_{\text{ref}}$:

$$\mathcal{L}_{\text{DPO}}(\pi_\theta; \pi_{\text{ref}}) = -\mathbb{E}_{(\mathcal{R}_w, \mathcal{R}_l) \sim \mathcal{D}} \left[ \log \sigma \left( \beta \log \frac{\pi_\theta(\mathcal{R}_w \mid I, q)}{\pi_{\text{ref}}(\mathcal{R}_w \mid I, q)} - \beta \log \frac{\pi_\theta(\mathcal{R}_l \mid I, q)}{\pi_{\text{ref}}(\mathcal{R}_l \mid I, q)} \right) \right], \quad (6)$$

where $\pi_\theta$ is the policy being trained, $\beta$ is a hyperparameter controlling the deviation from the reference policy, and $\sigma$ is the sigmoid function.

We employ an iterative online scheme: (1) Train $\pi_\theta$ on the current preference set $\mathcal{D}$ using Eq. 6. (2) Use the updated $\pi_\theta$ to sample new trajectories for the tasks (exploration). (3) Apply the preference criteria (1-3) to the new trajectories to generate an updated preference set $\mathcal{D}'$. (4) Set $\mathcal{D} \leftarrow \mathcal{D}'$ and $\pi_{\text{ref}} \leftarrow \pi_\theta$, then repeat from step (1). This iterative process enables the LRM to refine its tool usage strategy, continuously improving performance via on-policy interaction with the environment.

## 4 Experiments

### 4.1 Tasks and Datasets

We evaluate WebThinker on two primary task categories: **Complex Reasoning Benchmarks:** Tests multi-step reasoning with external knowledge using: **GPQA** [41] (PhD-level science QA in physics, chemistry, biology); **GAIA** [32] (general AI assistant evaluation on complex information acquisition tasks); **WebWalkerQA** [56] (deep web navigation and information extraction); and **Humanity's Last Exam (HLE)** [37] (extremely challenging problems across disciplines requiring advanced search and reasoning skills). Accuracy is judged by Qwen2.5-72B-Instruct [62]. **Scientific Report Generation:** Evaluates synthesis of research reports for open-ended questions using **glaiveai/reasoning-v1-20m**

Table 2: Main results on Humanity's Last Exam. We report Pass@1 metric for all tasks. For 32B models, the best results are in **bold** and the second are underlined. Results from larger or closed-sourced models are in gray color for reference. '†' denotes results from their official releases.

| Method | Humanity's Last Exam (Extremely Hard Reasoning Tasks) | | | | | | | | |
| | Math | Bio/Med | Physics | CS/AI | Human. | Chem. | Engineer. | Other | Avg. |
|---|---|---|---|---|---|---|---|---|---|
| *Direct Reasoning (w/o Retrieval)* | | | | | | | | | |
| Qwen2.5-32B | 6.0 | 7.0 | 2.0 | 3.2 | 10.0 | 0.0 | 5.3 | 4.4 | 5.4 |
| DeepSeek-R1-32B | 6.9 | 8.3 | 3.5 | 4.5 | 7.3 | 7.4 | 7.0 | 5.7 | 6.4 |
| QwQ-32B | 12.6 | 14.0 | 4.0 | 7.9 | 6.0 | 13.3 | 5.3 | 4.4 | 9.6 |
| GPT-4o† | 2.4 | 5.3 | 2.2 | 1.2 | 2.9 | 1.9 | 1.3 | 3.5 | 2.6 |
| Gemini-2.0-Flash-Thinking† | 8.5 | 7.4 | 5.3 | 5.8 | 7.1 | 6.5 | 3.8 | 4.0 | 7.1 |
| DeepSeek-R1-671B† | 9.3 | 8.6 | 5.8 | 7.4 | 11.0 | 5.6 | 10.3 | 7.5 | 8.6 |
| o3-mini (Medium)† | 14.0 | 9.8 | 11.5 | 8.2 | 6.7 | 10.2 | 7.7 | 6.5 | 11.1 |
| o3-mini (High)† | 18.8 | 11.1 | 14.2 | 11.1 | 6.2 | 10.2 | 7.7 | 8.0 | 14.0 |
| *Enhancing Reasoning with RAG Workflow* | | | | | | | | | |
| RAG-Qwen2.5-32B | 4.2 | 7.0 | **8.2** | 1.6 | 8.0 | 13.3 | 5.3 | 0.0 | 4.8 |
| w/ Query Planning | 6.0 | 4.7 | 4.0 | 6.3 | 4.0 | 6.7 | 5.3 | 6.7 | 5.6 |
| w/ Iterative RAG | 5.1 | 7.0 | 4.0 | 4.8 | 6.0 | 13.3 | 5.3 | 4.4 | 5.4 |
| RAG-QwQ-32B | 7.9 | 14.0 | 2.0 | 4.8 | 14.0 | 0.0 | 0.0 | 4.4 | 7.2 |
| w/ Query Planning | 11.2 | 16.3 | 4.0 | 4.8 | 12.0 | 6.7 | 0.0 | 11.1 | 9.6 |
| w/ Iterative RAG | 10.2 | 14.0 | 4.0 | 7.9 | 10.0 | 13.3 | 10.5 | 8.9 | 9.6 |
| *Autonomous Search within Reasoning* | | | | | | | | | |
| OpenAI Deep Research† | - | - | - | - | - | - | - | - | 26.6 |
| Search-o1-32B | 12.1 | 11.6 | 2.0 | 7.9 | 14.0 | 6.7 | 10.5 | **15.6** | 10.8 |
| WebThinker-32B-Base | 14.9 | 16.3 | 6.0 | 9.5 | 6.0 | 20.0 | **21.1** | **15.6** | 13.0 |
| WebThinker-32B-RL | **16.7** | **25.6** | 2.0 | **12.7** | **18.0** | **26.7** | 15.8 | **15.6** | **15.8** |

**(Glaive)** [11], a large-scale dataset with general, open-ended reasoning questions covering a wide range of subjects. Reports are evaluated using average scores judged by DeepSeek-R1-671B [14] and GPT-4o [19]. Details on datasets and evaluation are in Appendix B and C.

## 4.2 Baselines

We compare against three types of methods: **(1) Direct Reasoning:** Models using only internal knowledge without search, including open-source models (Qwen2.5-32B/72B-Instruct [62], Qwen2.5-Coder-32B-Instruct [18], QwQ-32B [48], Llama3.3-70B-Instruct [7]) and closed-source models (DeepSeek-R1-671B [14], GPT-4o [19], o1-preview [34], o3-mini [36], Gemini-2.0-Flash-Thinking [10]). **(2) Retrieval-Augmented Reasoning:** Methods using external knowledge from search engines: (i) Standard RAG (retrieves for the original query); (ii) RAG w/ Query Planning (decomposes query, retrieves for sub-queries, then generates); and (iii) Iterative RAG (retrieves information iteratively). **(3) Autonomous Search within Reasoning:** Systems integrating search actions into reasoning, including the open-source Search-o1 framework [26], and non-proprietary systems like OpenAI Deep Research [35], Grok3 DeeperSearch [12], and Gemini2.0 Deep Research [10].

## 4.3 Implementation Details

We use QwQ-32B [48] as WebThinker's backbone in our main results. Assistant models use Qwen2.5-Instruct [62] with the same parameters as the backbone. Generation uses max 81920 tokens, temperature 0.7, top_p 0.8, top_k 20, and repetition penalty 1.05. Search uses Bing Web Search API (US-EN region, k=10) with content fetched via Crawl4AI [49]. Training involves 2 iterations of online DPO with a max sequence length of 32,768. For baselines not trained for o1-like reasoning, we use Chain-of-Thought (CoT) [55] prompting. Detailed instructions can be found in Appendix C.

## 4.4 Results on Complex Problem-Solving

**Main Results.** Tables 1 and 2 show main results on complex reasoning tasks. Key findings include:

| Method | Glaive (General Research Tasks) | | | | |
| --- | --- | --- | --- | --- | --- |
| | Comp. | Thorough. | Fact. | Coherence | Avg. |
| *Retrieval-Augmented Report Generation* | | | | | |
| RAG-Qwen2.5-72B | 5.7 | 5.3 | 6.4 | 6.3 | 5.9 |
| RAG-DeepSeek-R1 | 6.6 | 6.4 | 7.1 | 7.1 | 6.8 |
| *Non-Proprietary Systems* | | | | | |
| Grok3 DeeperSearch | 6.4 | 6.1 | 7.0 | 6.5 | 6.5 |
| Gemini2.0 Deep Research | 8.1 | 8.0 | **7.7** | 7.7 | 7.9 |
| *Autonomous Think-Search-and-Draft (Ours)* | | | | | |
| WebThinker-32B-Base | **8.4** | 8.2 | **7.7** | 7.8 | 8.0 |
| WebThinker-32B-RL | 8.3 | **8.4** | **7.7** | **7.9** | **8.1** |

Analysis of Information Scopes

Figure 4: Main results on scientific report generation for general research tasks. **Left**: Overall performance comparison, reporting average scores evaluated by DeepSeek-R1 and GPT-4o. **Right**: t-SNE visualization of content embeddings from three randomly sampled topics.

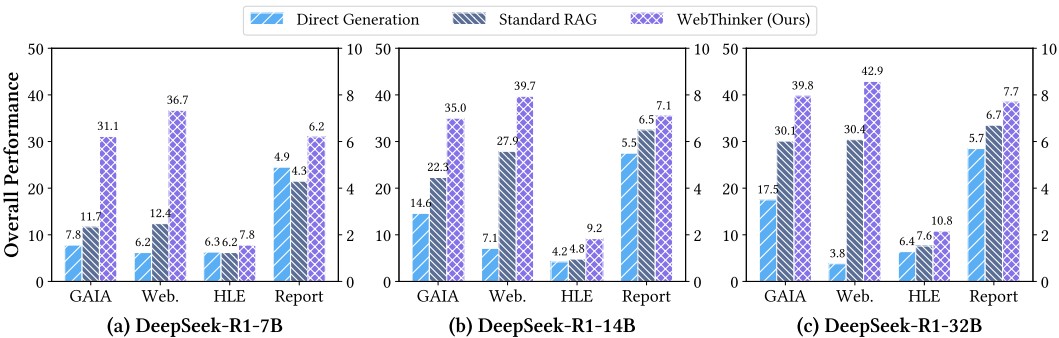

Figure 5: Analysis of the performance of WebThinker with DeepSeek-R1-based models across 7B, 14B, and 32B sizes, compared with direct generation and standard RAG approaches.

1. **Base LRM & RAG Workflow Limitations:** Reasoning models (e.g., QwQ-32B) surpass standard LLMs (e.g., Qwen2.5-72B on GPQA avg.) but falter on knowledge-intensive tasks (GAIA, WebWalkerQA). RAG improves performance on these tasks but shows inconsistent gains on complex HLE tasks that require deep integration of reasoning and search.

2. **Autonomous Search Advantage:** Autonomous search (e.g., Search-o1) yields notable gains over direct reasoning and basic RAG, especially on GAIA, WebWalkerQA, and HLE (e.g., HLE avg: Search-o1 10.8 vs. RAG-QwQ-32B 7.2).

3. **WebThinker Framework Superiority:** Our training-free WebThinker-32B-Base, utilizing its Deep Web Explorer for deeper web exploration, consistently surpasses prior methods like Search-o1 across all benchmarks (e.g., +22.9% on WebWalkerQA and +20.4% on HLE, respectively).

4. **RL Improvement:** RL-trained WebThinker-32B-RL achieves SOTA among 32B models on all benchmarks, substantially improving over its Base version (e.g., +8.5% on GAIA and +21.5% on HLE, respectively). Notably, on HLE, it surpasses even stronger models like o3-mini (High).

## 4.5 Results on Scientific Report Generation

**Main Results.** Figure 4 (left) presents WebThinker's performance on the Glaive scientific report generation task against RAG and non-proprietary baselines, evaluated on Completeness (Comp.), Thoroughness (Thorough.), Factuality (Fact.), and Coherence. **WebThinker achieves the top overall score (8.0), surpassing RAG baselines and advanced systems like Gemini-Deep Research (7.9).** It excels in Completeness (8.4) and Thoroughness (8.2), matching top Factuality (7.7) and Coherence (7.8) scores. **These results highlight the Autonomous Think-Search-and-Draft strategy's effectiveness in enabling LRMs to iteratively refine content through dynamic information gathering,** yielding more comprehensive and coherent reports than predefined RAG workflows.

Table 3: Ablation studies of WebThinker. **Left:** Performance on complex reasoning benchmarks (GAIA, WebWalkerQA, HLE). **Right:** Performance on the scientific report generation tasks (Glaive).

| Method | Complex Problem-Solving | | | | | Method | Scientific Report Generation | | | | |
|---|---|---|---|---|---|---|---|---|---|---|---|
| | GPQA | GAIA | Web. | HLE | Avg. | | Comp. | Tho. | Fact. | Coh. | Avg. |
| **WebThinker-32B-RL** | **70.7** | **48.5** | **46.5** | **15.8** | **45.4** | **WebThinker-32B-RL** | 8.3 | **8.4** | **7.7** | **7.9** | **8.1** |
| w/ Offline DPO | 69.2 | 45.6 | 44.0 | 14.2 | 43.2 | w/o Training (Base) | **8.4** | 8.2 | **7.7** | 7.8 | 8.0 |
| w/o Training (Base) | 68.7 | 44.7 | 41.9 | 13.0 | 42.1 | w/o Deep Web Explorer | 7.9 | 7.9 | 7.5 | 7.6 | 7.7 |
| w/o Link Clicking | 69.7 | 42.7 | 42.6 | 15.2 | 42.6 | w/o Report Check & Edit | 8.1 | 8.0 | 7.6 | 6.9 | 7.7 |
| w/o Deep Web Explorer | 63.6 | 38.8 | 38.5 | 12.0 | 38.3 | w/o Auto. Report Draft | 6.3 | 6.4 | 6.8 | 7.0 | 6.6 |

**Analysis of Information Scopes.** Qualitative t-SNE visualization of content embeddings for reports on three randomly sampled Glaive topics (Figure 4, right) shows distinct topic clusters, where each point is a report's embedding. **Notably, WebThinker's reports often form broader sub-clusters within each topic group. This suggests WebThinker, with its Deep Web Explorer and iterative drafting, explores and synthesizes information from more diverse perspectives and depths than other methods.** Its autonomous nature allows dynamic adaptation of information gathering and writing to evolving report needs, yielding richer and more unique information coverage.

## 4.6 Adaptability of WebThinker Across Different LRM Backbones.

We assessed WebThinker's adaptability with DeepSeek-R1 models of varying sizes (7B, 14B, 32B; Figure 5). To ensure stable tool usage, we conducted cold-start SFT (using 7.8k trajectories from QwQ-32B WebThinker) followed by RL training (Section 3.5). On all tasks and model sizes, R1-based WebThinker consistently outperformed direct reasoning and standard RAG. For example, **WebThinker-R1-7B achieved relative gains of 174.4% (GAIA) and 422.6% (WebWalkerQA) over direct generation, and 82.9% (GAIA) and 161.3% (WebWalkerQA) over standard RAG.** Similar substantial improvements were observed across other settings, demonstrating WebThinker's general applicability and effectiveness in enhancing diverse LRMs' deep research capabilities.

## 4.7 Ablation Studies

Ablation studies (Table 3) assessed key WebThinker components. **(1) RL Training:** Iterative online RL markedly improves problem-solving (Avg. 44.9 vs. 42.1 Base, 43.2 offline DPO), validating our training strategy. Its impact on report generation is minimal, as the base framework is already effective. **(2) Deep Web Exploration:** Removing the Deep Web Explorer severely degrades both problem-solving (Avg. 38.3) and report generation (Avg. 7.7), proving its criticality. Disabling link-clicking alone also impairs problem-solving (Avg. 42.6), highlighting the value of deeper exploration. **(3) Report Generation Components:** Removing autonomous drafting causes the largest quality drop in report generation (Avg. 6.6), stressing the importance of interleaved thinking, searching, and drafting. Disabling check-and-edit tools also reduces quality (Avg. 7.7), notably coherence (6.9 vs. 7.9), reinforcing the need for iterative refinement. These findings affirm WebThinker's design.

## 5 Conclusion

In this work, we present WebThinker, which equips large reasoning models with deep research capabilities by enabling autonomous web exploration and comprehensive report generation through agentic reasoning. We develop: (1) a Deep Web Explorer for dynamic web page navigation, (2) an autonomous Think-Search-and-Draft strategy that integrates reasoning, searching, and writing, and (3) RL-based training strategies to enhance research tool utilization. Experiments on complex reasoning benchmarks and scientific report generation tasks demonstrate that WebThinker outperforms existing methods and proprietary systems. Despite its strengths, WebThinker has several limitations that require further improvement. First, it cannot process multimodal information such as images and videos, making the development of **multimodal deep research** systems an important direction. Second, it currently supports only a limited set of tools, underscoring the need for **tool scalability and generalization**. Finally, extending WebThinker to support **GUI-based web exploration** would enable it to handle more complex and real-world interactive tasks.

## Acknowledgments

This work was supported by Beijing Municipal Science and Technology Project No. Z231100010323009, National Natural Science Foundation of China No. 62272467, Beijing Natural Science Foundation No. L233008. The work was partially done at the Engineering Research Center of Next-Generation Intelligent Search and Recommendation, MOE.

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

# Appendix

# A    Inference Process

## A.1    Research-Related Tools

WebThinker utilizes several tools during its inference process:

- **Search Tool:** This tool is employed by both the main reasoning process and the Deep Web Explorer. In the main process, it invokes the Deep Web Explorer, while within the Explorer, it calls a standard search engine API. The model signals a search query by generating `<|begin_search_query|>` and `<|end_search_query|>`. The processed search results are then returned within `<|begin_search_result|>` and `<|end_search_result|>`.

- **Link & Button Clicking Tool:** This tool is exclusively used by the Deep Web Explorer to interact with web pages, typically invoking a web crawler. We use Crawl4AI [49] for fetching web page content. The model triggers this tool by generating `<|begin_click_link|>` and `<|end_click_link|>`. The content retrieved from the clicked link (potentially summarized) is returned within `<|begin_click_result|>` and `<|end_click_result|>`.

- **Report Writing Tools:** These tools are specific to the Report Generation Mode and are invoked by the main reasoning process. They facilitate drafting and refining the research report.
    - **Section Writing:** Triggered by `<|begin_write_section|>` section_name\ncontents_to_write `<|end_write_section|>`. This command instructs an assistant LLM to write the specified section based on the provided content guidelines and gathered information. It does not return content directly into the main reasoning flow to maintain coherence.
    - **Report Checking:** Triggered by `<|begin_check_article|>`. This command requests the current outline of the report. The system extracts all section and subsection titles to form the outline, which is then inserted into the reasoning context followed by `<|end_check_article|>`. This avoids injecting the full report text, which could disrupt the reasoning flow.
    - **Report Editing:** Triggered by `<|begin_edit_article|>` edit_instruction `<|end_edit_article|>`. This command instructs an assistant LLM to modify the existing report based on the provided editing instructions. Similar to section writing, it doesn't return content directly into the main flow.

## A.2    Inference Process of WebThinker

The overall inference process in WebThinker involves two main components: the main reasoning process and the Deep Web Explorer for in-depth information gathering. As outlined in Section 3.2, the main reasoning process operates in two distinct modes: Problem-Solving Mode and Report Generation Mode. Detailed instructions for each component and mode can be found in Section C.

### A.2.1    Inference for Main Reasoning Process

**Problem-Solving Mode:**    In this mode, the LRM receives a user's question and aims to find a direct answer. It primarily utilizes the **Search Tool**. When the model generates the `<|end_search_query|>` token, the generation process pauses. The system extracts the most recent query enclosed within `<|begin_search_query|>` and `<|end_search_query|>`. Concurrently, an assistant LLM formulates a detailed *search intent* based on the query and the preceding reasoning chain. The system then uses a web crawler to fetch the content of the top-k (k=10 in our experiments) retrieved web documents. The original query, the generated search intent, and the fetched web content are passed to the **Deep Web Explorer**. The Explorer processes this information and returns a refined summary relevant to the query and intent, enclosed within `<|begin_search_result|>` and `<|end_search_result|>`. The main LRM then resumes its reasoning with this new information. This cycle continues until the model generates an End-of-Sequence (EOS) token or reaches the maximum token limit. The final answer is extracted based on a predefined answer pattern (e.g., `\boxed{ANSWER}` or `ANSWER: {ANSWER}`).

**Report Generation Mode:**    This mode handles tasks requiring the generation of a comprehensive research report. It receives the user's question along with a research plan (potentially generated by an assistant LLM). The LRM utilizes the **Search Tool** (invoking the Deep Web Explorer as in Problem-Solving Mode) and the three **Report Writing Tools** (*Section Writing*, *Report Checking*, *Report Editing*). Similar to the problem-solving mode, generation pauses whenever the model outputs

a closing tool token (`<|end_search_query|>`, `<|end_write_section|>`, `<|end_edit_article|>`) or the opening token for checking (`<|begin_check_article|>`).

- For **search**, the process mirrors the Problem-Solving Mode: invoke the Deep Web Explorer and insert results between `<|begin_search_result|>` and `<|end_search_result|>`.

- For **section writing** and **report editing**, the corresponding assistant LLM performs the action in the background. Since no direct output is inserted, the main LRM continues its reasoning immediately after generating the end token.

- For **report checking**, the system generates the report outline, appends the `<|end_check_article|>` token, inserts this into the context, and resumes generation.

The model continues this loop of reasoning, searching, and drafting/editing until it generates an EOS token or reaches the maximum token limit. The final, complete report is the accumulated result of the writing and editing actions performed by the assistant LLM throughout the process.

### A.2.2 Inference for Deep Web Explorer

**Deep Web Explorer:** The Deep Web Explorer operates as a sub-process invoked by the main reasoning process when a search is needed. Its goal is to deeply investigate the initial search results provided by the main process. It has access to two tools: a direct **Search Engine** call and the **Link & Button Clicking Tool**. When the Explorer model generates `<|end_search_query|>` or `<|end_click_link|>`, generation pauses.

- A **search** action directly calls the search engine API (e.g., Bing) and returns the top-10 results, typically including titles, URLs, and snippets (not the full page content). These results are inserted into the Explorer's reasoning context.

- A **link click** action uses the crawler (Crawl4AI) to fetch the full content of the specified URL. To manage potentially long web pages, an assistant LLM summarizes the fetched content based on a generated "click intent", ensuring the inserted information is concise and relevant. The summarized result is placed within `<|begin_click_result|>` and `<|end_click_result|>`.

The Explorer continues its internal reasoning process—analyzing results, deciding whether to perform another search, or clicking on a promising link—until it gathers sufficient information relevant to the initial query and search intent provided by the main reasoning process. It concludes by generating its findings within a designated "Final Information" pattern, which is then passed back to the main reasoning process as the output within `<|begin_search_result|>` and `<|end_search_result|>`. This process stops when the Explorer generates an EOS token or reaches its maximum token limit.

## B  Datasets

### B.1  Complex Reasoning Tasks

For testing, we use the following four widely used datasets:

- **GPQA [41]:** Written by experts (holding or pursuing PhDs in relevant fields) in biology, physics, and chemistry, these "Google-proof" questions test the model's expertise and reasoning in these specific domains. It consists of multiple-choice questions (4 options each); we use the diamond set, totaling 198 questions.

- **GAIA [32]:** A benchmark dataset designed to evaluate the capabilities of general artificial intelligence (AGI). Compared to GPQA, GAIA focuses more on generality, testing AI performance across diverse tasks such as question answering, reasoning, code generation, and multimodal processing. As our model is text-based and cannot handle other modalities, we use the text-only validation subset, comprising 103 questions.

- **WebWalkerQA [56]:** A benchmark specifically designed to evaluate the web traversal capabilities of LLMs, simulating human ability to explore websites via clicking to find required information. It includes 680 challenging queries covering multilingual and multi-domain web content. We use the test set for evaluation, totaling 680 questions.

- **Humanity's Last Exam (HLE) [37]:** Aims to assess LLMs' capabilities at the frontiers of human knowledge. It contains 2500 challenging cross-disciplinary questions covering fields like mathematics, humanities, and natural sciences. The benchmark includes multiple-choice and short-answer questions, all with clear and easily verifiable solutions. Current state-of-the-art models achieve less than 10% accuracy on HLE, highlighting its difficulty and effectiveness in measuring advanced academic capabilities. Due to its large test set, we randomly sample 500 text-only questions for testing.

For training, we use the following datasets. We sample approximately 3k data points from these datasets. First, we perform direct generation using QwQ-32B and retain only the questions that cannot be answered correctly through direct generation. Then, for each of these questions, we generate three responses using WebThinker-32B-Base to filter for high-quality preference training data, following the criteria outlined in Section 3.5.

- **SuperGPQA [46]:** Designed to evaluate LLM knowledge and reasoning across 285 graduate-level subjects. It contains 26,529 multiple-choice questions covering 72 fields, grouped into 13 broader disciplines, with a strong emphasis on STEM subjects (77.2%).
- **WebWalkerQA (Silver) [56]:** We use the Silver set of WebWalkerQA, containing 13.6k data points. The description is the same as above.
- **OpenThoughts [47]:** Contains 114,000 carefully crafted reasoning examples designed to train AI models for complex logical and mathematical reasoning tasks. Given that we already use NuminaMath as training data, we prioritize sampling data related to STEM and puzzles.
- **NaturalReasoning [65]:** Aims to enhance LLM reasoning capabilities through 2.8 million challenging reasoning problems across multiple domains, including STEM (e.g., physics and computer science), economics, and social sciences.
- **NuminaMath [25]:** Features problems ranging in difficulty from high school exercises to International Mathematical Olympiad questions, sourced from various online platforms and PDFs. NuminaMath provides high-quality, structured data enabling AI models to learn and replicate expert-level mathematical reasoning.

### B.2 Scientific Report Generation Tasks

For the scientific report generation task, we use **glaiveai/reasoning-v1-20m (Glaive) [11]** for both training and testing. This dataset is a large-scale synthetic collection containing over 22 million general reasoning questions and responses generated using DeepSeek-R1-Distill-Llama-70B [14].

It aims to cover diverse non-code/math topics such as social and natural sciences, education, and creative writing. We sample 1.5k questions for each iteration's preference data construction and 30 questions for testing.

## C Instruction Templates

### C.1 Instructions for WebThinker

---

**Problem Solving Instruction for WebThinker**

You are a reasoning assistant with the ability to perform web searches to help you answer the user's question accurately. You have special tools:
- To perform a search: write <|begin_search_query|> your query here <|end_search_query|>.
Then, the system will search and analyze relevant web pages, then provide you with helpful information in the format <|begin_search_result|> ...search results... <|end_search_result|>.
You can repeat the search process multiple times if necessary.
Once you have all the information you need, continue your reasoning.
Example:
Question: "Alice David is the voice of Lara Croft in a video game developed by which company?"
Assistant thinking steps:
- I need to find out who voices Lara Croft in the video game.
- Then, I need to determine which company developed that video game.
Assistant:

---

<|begin_search_query|>Alice David Lara Croft voice<|end_search_query|>
(System returns processed information from relevant web pages)
Assistant thinks: The search results indicate that Alice David is the voice of Lara Croft in a specific video game. Now, I need to find out which company developed that game.
Assistant:
<|begin_search_query|>video game developed by Alice David Lara Croft<|end_search_query|>
(System returns processed information from relevant web pages)
Assistant continues reasoning with the new information...
Remember:
- Use <|begin_search_query|> to request a web search and end with <|end_search_query|>.
- When done searching, continue your reasoning.

---

## Instruction for Deep Web Explorer

You are a web explorer analyzing search results to find relevant information based on a given search query and search intent.
**Guidelines:**
1. **Analyze the Searched Web Pages:**
- Carefully review the content of each searched web page.
- Identify factual information that is relevant to the **Current Search Query** and can aid in the reasoning process for the original question.
2. **More Information Seeking:**
- If the information is not relevant to the query, you could:
1. Search again: <|begin_search_query|>another search query<|end_search_query|>
2. Access webpage content using: <|begin_click_link|>your URL<|end_click_link|>
3. **Extract Relevant Information:**
- Return the relevant information from the **Searched Web Pages** that is relevant to the **Current Search Query**.
4. **Output Format:**
- Present the information beginning with **Final Information** as shown below.
**Final Information**
[Relevant information]
**Inputs:**
- **Current Search Query:**
{search_query}
- **Detailed Search Intent:**
{search_intent}
- **Searched Web Pages:**
{search_result}
Now please analyze the web pages and extract relevant information for the search query "{search_query}" and the search intent.

---

## Report Generation Instruction for WebThinker

You are a research assistant with the ability to perform web searches to write a scientific research article. You have special tools:
- To perform a search: write <|begin_search_query|> your query here <|end_search_query|>.
Then, the system will search and analyze relevant web pages, then provide you with helpful information in the format <|begin_search_result|>search results<|end_search_result|>.
- To write a section of the research article: write <|begin_write_section|>section name\ncontents to write<|end_write_section|>.
Then, the system will completely write the section based on your request and current gathered information.
- To check the current article: write <|begin_check_article|>system returns outline of all current written contents<|end_check_article|>.
- To edit the article: write <|begin_edit_article|>your detailed edit goal and instruction<|end_edit_article|>.
Then, the system will edit the article based on your goal and instruction and current gathered information.
Your task is to research and write a scientific article about:
{question}
Here is a research plan to guide your investigation:
{plan}

Please follow the research plan step by step:
1. Use web searches to gather detailed information for each point
2. After each search, analyze the results and determine what additional information is needed
3. When you have sufficient information for a section, request to write that section
4. Continue this process until the full article is complete
5. Check the current article and edit sections as needed to improve clarity and completeness
Example:
`<|begin_search_query|>`first search query`<|end_search_query|>`
`<|begin_search_result|>`Summary of information from searched web pages`<|end_search_result|>`
Based on these results, I understand X, but still need to investigate Y...
`<|begin_search_query|>`follow-up search query focusing on Y`<|end_search_query|>`
`<|begin_search_result|>`Summary of information from searched web pages`<|end_search_result|>`
Now I have enough information to write the first section...
`<|begin_write_section|>`Introduction\nThis section should introduce ... `<|end_write_section|>`
I have written the introduction. Now I need to explore more information to write the next section ...
After writing the above sections, I need to check the current article to ensure the content is complete and accurate.
`<|begin_check_article|>`System returns outline of current written article`<|end_check_article|>`
Wait, I realize that I need to edit ...
`<|begin_edit_article|>`your edit instruction`<|end_edit_article|>`
Assistant continues gathering information and writing sections until getting comprehensive information and finishing the entire article.
Remember:
- Use `<|begin_search_query|>`query`<|end_search_query|>` to get information from web searches
- Use `<|begin_write_section|>`section name\ncontents to write`<|end_write_section|>` to call the system to write a section in the article
- Use `<|begin_check_article|>`outline of current article`<|end_check_article|>` to check the current written article
- Use `<|begin_edit_article|>`edit instruction`<|end_edit_article|>` to call the system to edit and improve the article
- You should strictly follow the above format to call the functions.
- Do not propose methods or design experiments, your task is to comprehensively research with web searches.
- Do not omit any key points in the article.
- When you think the article is complete, directly output "I have finished my work." and stop.
Now begin your research and write the article about:
`{question}`

---

### Instruction for Writing Section

You are a research paper writing assistant. Please write a complete and comprehensive "`{section_name}`" section based on the following information.
**Potential helpful documents:**
`{relevant_documents}`
**Original question:**
`{question}`
**Previous thoughts:**
`{previous_thoughts}`
**Outline of current written article:**
`{current_article}`
**Name of the next section to write:**
## `{section_name}`
**Your task is to comprehensively write the next section based on the following goal:**
`{task}`
**Note:**
- Write focused content that aligns with the above goal for this section.
- No need to mention citations or references.
- Each paragraph should be comprehensive and well-developed to thoroughly explore the topic. Avoid very brief paragraphs that lack sufficient detail and depth.
- If possible, add markdown tables to present more complete and structured information to users.
Please provide the comprehensive content of the section in markdown format. ## `{section_name}`

**Instruction for Editing Article**

You are a professional article editor. Please help me modify the article based on the following edit instruction:
**Edit instruction:**
{edit_instruction}
**Current article:**
{article}
Please output the complete modified article incorporating all the requested changes.
**Note:**
- Keep all original content that doesn't need modification. (Do not just output the modified content, but output the entire modified article.)
- Make all edits specified in the edit instructions.
- Output format:

```markdown
...
```

Please provide the complete modified article in markdown format.

---

**Instruction for Search Plan Generation**

Please help me create a detailed plan to search over the web for solving the following question:
{query}
Your task is to comprehensively gather all relevant information to thoroughly solve the user's question.
Note:
- No need to mention citations or references.
- Do not propose methods or design experiments, your task is to research user's question with web searches.
- Be comprehensive and thorough, do not miss any relevant information.
- No more than 8 steps.
Please output the plan in numbered steps like:
(1) ...
(2) ...
etc.
Directly output the plan, do not include any other words.

---

**Instruction for Search Intent Generation**

Based on the previous thoughts below, provide the detailed intent of the latest search query.
Previous thoughts:
{previous_thoughts}
Please provide the current search intent.

---

**Instruction for Click Intent Generation**

Based on the previous thoughts below, provide the detailed intent of the latest click action.
Original question:
{question}
Previous thoughts:
{prev_reasoning}
Please provide the current click intent.

## C.2 Task Instructions

The task instruction specifies the description of a specific task and the answer output format for a particular model. These instructions are directly concatenated after the method instructions (such as those for WebThinker, Search-o1, or RAG approaches).

### C.2.1  Task Instruction for QwQ-based Models

---
**Task Instruction for QwQ-based Models**

Please answer the following question.   You should provide your final answer in the format
\boxed{YOUR_ANSWER}.
Question:
`{question}`

---

### C.2.2  Task Instruction for DeepSeek-R1-based Models

---
**Task Instruction for DeepSeek-R1-based Models**

Please answer the following question.
Provide your final answer in the format **ANSWER: {YOUR_ANSWER}**.
Question:
`{question}`

---

## C.3  Instructions for Evaluation

In this work, we use LLM-as-Judges to evaluate both complex problem-solving and scientific report generation tasks. The specific instructions are as follows:

### C.3.1  Evaluation Instruction for Problem-Solving Tasks

We use Qwen2.5-72B-Instruct [62] to evaluate all complex problem-solving tasks. An output labeled as "Correct" is considered correct, while "Incorrect" is considered wrong. In cases where the predicted answer cannot be accurately extracted, we directly use the last five lines of the model's output as the predicted answer. This approach helps reduce evaluation inaccuracies caused by formatting issues, case sensitivity, and similar factors, aligning with the official evaluation methods of benchmarks like WebWalkerQA [56] and HLE [37].

---
**Evaluation Instruction for Problem-Solving Tasks**

You are an evaluation assistant. Please determine if the predicted answer is equivalent to the labeled answer.
Question:
`{question}`
Labeled Answer:
`{labeled_answer}`
Predicted Answer:
`{pred_answer}`
Are these answers equivalent? Please respond with "Correct" if they are equivalent, or "Incorrect" if they are not
equivalent. Do not include any other text.

---

### C.3.2  Evaluation Instruction for Report Quality

For scientific report generation tasks, we use DeepSeek-R1-671B [14] and GPT-4o [19] respectively with the following instruction for evaluation, obtain scores, and take their average. Here, `{system_a}` to `{system_e}` are sequentially the reports generated by the systems being evaluated. We perform listwise evaluation to better compare the quality differences between reports generated by different systems. The input order is randomized rather than fixed to reduce bias caused by the model's context window. We did not employ human evaluation because the reports generated by each system have distinct features, making it easy for humans to identify their originating system, which introduces significant bias. Model-based evaluation is therefore more impartial.

Research Question:
{question}
Please objectively evaluate the quality of research articles generated by systems A, B, C, D and E for this question, and provide scores out of 10 for the following criteria:
(1) Overall Comprehensiveness: The report should cover content as comprehensively as possible
(2) Thoroughness of Discussion: Each section should be discussed thoroughly, not just superficially
(3) Factuality: There should be minimal factual errors
(4) Coherence: The discussion should stay focused and relevant to the topic
Notes:
- A satisfactory performance deserves around 5 points, with higher scores for excellence and lower scores for deficiencies
- You should not easily assign scores higher than 8 or lower than 3 unless you provide substantial reasoning.
- You do not need to consider citations in the articles
——————————————————— Research article generated by system A: ———————————————————
{system_a}
———————————————————————————————————————————————————————
——————————————————— Research article generated by system B: ———————————————————
{system_b}
———————————————————————————————————————————————————————
——————————————————— Research article generated by system C: ———————————————————
{system_c}
———————————————————————————————————————————————————————
——————————————————— Research article generated by system D: ———————————————————
{system_d}
———————————————————————————————————————————————————————
——————————————————— Research article generated by system E: ———————————————————
{system_e}
———————————————————————————————————————————————————————

Research Question:
{question}
Please objectively evaluate the quality of research articles generated by systems A, B, C, D and E for this question, and provide scores out of 10 for the following criteria:
(1) Overall Comprehensiveness: The report should cover content as comprehensively as possible
(2) Thoroughness of Discussion: Each section should be discussed thoroughly, not just superficially
(3) Factuality: There should be minimal factual errors
(4) Coherence: The discussion should stay focused and relevant to the topic
Notes:
- A satisfactory performance deserves around 5 points, with higher scores for excellence and lower scores for deficiencies
- You should not easily assign scores higher than 8 or lower than 3 unless you provide substantial reasoning.
- You do not need to consider citations in the articles
Please analyze each article and provide the final scores in the following JSON format:

```json
{
  "System A": {
    "Overall Comprehensiveness": ,
    "Thoroughness of Discussion": ,
    "Factuality": ,
    "Coherence":
  },
  "System B": {
    "Overall Comprehensiveness": ,
    "Thoroughness of Discussion": ,
    "Factuality": ,
    "Coherence":
  },
```

```
  "System C": {
    "Overall Comprehensiveness": ,
    "Thoroughness of Discussion": ,
    "Factuality": ,
    "Coherence":
  },
  "System D": {
    "Overall Comprehensiveness": ,
    "Thoroughness of Discussion": ,
    "Factuality": ,
    "Coherence":
  },
  "System E": {
    "Overall Comprehensiveness": ,
    "Thoroughness of Discussion": ,
    "Factuality": ,
    "Coherence":
  }
}
```

### C.4 Additional Notes

All instructions presented above are provided as user prompts rather than system prompts. When applying the WebThinker approach (detailed in Section C.1) to models based on QwQ-32B [48] or DeepSeek-R1 series [14], the corresponding task instruction (from Section C.2.1 or C.2.2) is appended after the main WebThinker instruction. For other models, such as Qwen2.5-32B-Instruct [62], Qwen2.5-72B-Instruct [62], and GPT-4o [19], we include the Chain-of-Thought prompt "You should think step by step to solve it." before the question to explicitly encourage step-by-step reasoning prior to providing the final answer, following [55].

For detailed instructions regarding the baseline methods like Search-o1, Standard RAG, etc., you can refer to the appendix of Search-o1 [26] or visit our WebThinker GitHub repository: https://github.com/RUC-NLPIR/WebThinker.

## D Case Study

The examples from Tables 4, 5, 6, 7, 8, 9, and 10 demonstrate WebThinker's effectiveness across different capabilities including complex problem solving, deep web exploration, and scientific report generation.

### D.1 Case Study for Complex Problem Solving

WebThinker shows strong reasoning capabilities in complex problem-solving scenarios:

- In the GAIA dataset example (Table 4), WebThinker correctly identifies that Nemo from "Finding Nemo" is a clownfish, and methodically searches for nonnative sightings in the USGS database, finding the only documented case at Fred Howard Park, Florida. It then determines the correct zip code (34689).

- For the WebWalkerQA example (Table 5), WebThinker resolves ambiguity about the "evening after" the ACL 2023 awards ceremony by searching for relevant dates and determining that the social event occurred on the same day (July 11) from 7:00 PM to 10:30 PM, not the following day.

- In the mathematical problem from Humanity's Last Exam (Table 6), WebThinker demonstrates sophisticated reasoning by finding the formula for simplicial volume of surfaces and their products, correctly calculating the simplicial volume of $\Sigma_{31} \times \Sigma_{17}$ as 11520.

### D.2 Case Study for Deep Web Explorer

The Deep Web Explorer demonstrates effective information retrieval and integration:

- For the ASH Annual Meeting deadlines (Table 7), the explorer not only finds the late-breaking abstract submission dates but also discovers the ancillary meeting deadline by clicking on a PDF link, compiling comprehensive deadline information with specific dates and requirements.
- In the CLTS and Aedes mosquito control example (Table 8), the explorer clicks on a repository link to find a case study integrating Community-Led Total Sanitation with mosquito control in Indonesia, Vietnam, and the Philippines, providing specific outcomes (40% reduction in breeding sites).

### D.3 Case Study for Scientific Report Generation

WebThinker's report generation capability is illustrated in Tables 9 and 10:

- The model systematically follows a research plan for optimizing 3D printed lattice structures, conducting searches on FDM limitations, lattice optimization techniques, and material properties.
- It methodically writes, checks, and edits sections, addressing duplicate content and ensuring coherent structure.
- The final report (Table 10) presents a comprehensive analysis of lattice optimization for robotic nodes, covering technical aspects from printing limitations to material selection, with practical applications and case studies.

These examples validate WebThinker's effectiveness across complex problem-solving, deep web exploration, and scientific report generation tasks.

## E  Summary of Contributions

In summary, the core contributions of this paper is as follows:

1. We introduce WebThinker, a deep research agent that **autonomously search, deeply explore web pages,** and **draft research reports**, all within its thinking process. Unlike traditional predefined workflow, WebThinker enables the LRM itself to perform actions on its own while thinking, achieving end-to-end task execution in a single generation.
2. We propose a **Deep Web Explorer** that empowers LRMs with web search and navigation capabilities to deeply gather, traverse, and extract high-quality information from the web.
3. We introduce an **Autonomous Think-Search-and-Draft** strategy that enables real-time report writing during the thinking and searching process.
4. We develop **RL-based training strategies** that iteratively synthesize tool-usage preference data and apply online DPO training to enhance the LRM's tool utilization capabilities.
5. Extensive experiments demonstrate the effectiveness of WebThinker on **complex reasoning tasks** and **scientific report generation tasks** with both QwQ-based [48] and DeepSeek-R1-based [14] LRM backbones.

## F  Broader Impact

This work introduces WebThinker, a framework designed to significantly enhance the deep research capabilities of Large Reasoning Models (LRMs). By empowering LRMs to autonomously explore the web, synthesize information from diverse sources, and generate comprehensive reports, WebThinker addresses critical limitations in tackling knowledge-intensive real-world tasks. The potential broader impact of our research is considerable, offering a new paradigm for how complex information is accessed, processed, and utilized across various domains. This could accelerate scientific discovery, improve the quality of investigative journalism, and support more informed decision-making in sectors like finance and policy-making by providing powerful tools for in-depth research. Moreover, WebThinker has the potential to democratize access to advanced research capabilities, fostering innovation and learning.

The societal consequences of deploying such advanced research agents demand careful and proactive consideration. The ability to autonomously gather, interpret, and synthesize vast quantities of online

information carries risks, including the potential for generating and disseminating sophisticated misinformation, or inadvertently amplifying biases present in the training data and web sources. Concerns regarding data privacy, the verifiability of AI-generated research, and the potential displacement of human expertise in research-intensive roles must also be addressed. Therefore, it is imperative that the advancement of technologies like WebThinker is coupled with the development of ethical guidelines, robust validation mechanisms, and a commitment to transparency to mitigate potential harms and ensure these tools serve the broader public good.

Table 4: A **problem-solving** example from **main reasoning process** of WebThinker on GAIA dataset, with special symbols used in the search queries and search results highlighted in `blue box`. [...] indicates omitted content.

---

**Example # from GAIA Dataset**

**Question:**
I'm researching species that became invasive after people who kept them as pets released them. There's a certain species of fish that was popularized as a pet by being the main character of the movie Finding Nemo. According to the USGS, where was this fish found as a nonnative species, before the year 2020? I need the answer formatted as the five-digit zip codes of the places the species was found, separated by commas if there is more than one place.

**Labeled Answer:**
34689

---

**Model Output by WebThinker**

Okay, let's tackle this question step by step. The user wants to know the five-digit zip codes of places in the US where the clownfish (the main character from Finding Nemo) has been found as a nonnative species according to the USGS, before 2020.

First, I need to confirm which species is the main character in Finding Nemo. That's Nemo the clownfish, specifically *Amphiprion ocellaris*. It's important to look for the correct species.

Next, the user mentioned the USGS database on nonindigenous aquatic species. I need to check this database for records before 2020.

The task requires the answer formatted as five-digit zip codes. I need to find out if the USGS data provides this level of detail or if I'll need to find the zip codes based on reported locations (like cities or parks).

Let me start by searching the USGS for nonnative sightings of *Amphiprion ocellaris* before 2020.

`<|begin_search_query|>` USGS Amphiprion ocellaris nonnative sightings pre 2020 `<|end_search_query|>`

`<|begin_search_result|>` The only documented nonnative sighting of *Amphiprion ocellaris* (clown anemonefish) in the United States prior to 2020, as reported by the USGS Nonindigenous Aquatic Species (NAS) database, occurred in **Pinellas County, Florida**, specifically at **Fred Howard Park** in the Gulf of Mexico region during **2018**. [...] The provided web pages do not explicitly mention the **five-digit zip code** associated with this location. [...] For precise zip code details, accessing the full NAS database record [...] may be necessary. `<|end_search_result|>`

Okay, the search result indicates only one sighting: Fred Howard Park in Pinellas County, Florida, in 2018. It explicitly states the zip code isn't provided in the search results.

So, I need to find the zip code for Fred Howard Park. The result mentions Pinellas County, but the park might be in a specific city within the county. Let me search for the park's zip code.

`<|begin_search_query|>` Fred Howard Park St. Petersburg Florida zip code `<|end_search_query|>`

`<|begin_search_result|>` The zip code for Fred Howard Park in Tarpon Springs, Florida (part of Pinellas County) is **34689**. While the search query references St. Petersburg, the park is officially located in Tarpon Springs [...]. The address consistently listed [...] is **1700 Sunset Drive, Tarpon Springs, FL 34689**. [...] `<|end_search_result|>`

Ah, the park is in Tarpon Springs, FL, within Pinellas County, and the zip code is 34689.

The first search result mentioned this was the *only* documented sighting before 2020. To be absolutely sure, I'll run the first search query again to confirm no other locations were missed.

`<|begin_search_query|>` Amphiprion ocellaris USGS nonnative sightings before 2020 `<|end_search_query|>`

`<|begin_search_result|>` The USGS Nonindigenous Aquatic Species (NAS) database indicates that *Amphiprion ocellaris* (clown anemonefish) has only one documented nonnative sighting in the United States prior to 2020. This sighting occurred in **Fred Howard Park**, **Pinellas County**, **Florida**, in **2018**. [...] No other locations besides Fred Howard Park are listed [...] as having nonnative sightings of *A. ocellaris* before 2020. The five-digit zip code for Fred Howard Park, Tarpon Springs, Florida, is **34689**. [...] `<|end_search_result|>`

Confirmed again. The only documented nonnative sighting before 2020 was at Fred Howard Park, zip code 34689. Since the user asked for zip codes separated by commas if there's more than one, and there's only one, the answer is just this single zip code.

**Final Answer**

34689

---

Table 5: A **problem-solving** example from **main reasoning process** of WebThinker on WebWalkerQA dataset, with special symbols used in the search queries and search results highlighted in `blue box`. [...] indicates omitted content.

---

**Example from WebWalkerQA Dataset**

---

**Question:**
What was the specific schedule for the social event held on the evening after the ACL 2023 best paper awards ceremony?

**Labeled Answer:**
The social event was scheduled from 7:00 PM to 10:30 PM on July 11th.

---

**Model Output by WebThinker**

---

Okay, so the user is asking about the specific schedule of a social event that took place on the evening *after* the ACL 2023 best paper awards ceremony. First, I need to figure out when the awards ceremony actually happened. Let me search for that specifically.

`<|begin_search_query|>` ACL 2023 best paper awards ceremony date `<|end_search_query|>`

`<|begin_search_result|>` The ACL 2023 best paper awards ceremony took place on **July 11, 2023**, according to multiple sources: [...] Web Page 4 [...] explicitly states that the awards were announced on July 11, 2023. [...] Web Page 5 [...] mentions the award presentation occurred on July 11, 2023 [...]. Web Page 6 [...] confirms the awards were announced on July 11, 2023. [...] Web Page 9 [...] also indicates the awards were announced on July 11, 2023. [...] the awards ceremony itself was held on the **11th**. [...] `<|end_search_result|>`

Okay, the awards ceremony was definitively on July 11, 2023. The question asks for the social event on the evening *after* the ceremony, which would be the evening of July 12. Let me search for the social events schedule to see what was planned.

`<|begin_search_query|>` ACL 2023 social events schedule `<|end_search_query|>`

`<|begin_search_result|>` The ACL 2023 social events schedule includes the following relevant details: 1. **Social Event on July 11, 2023**: - **Time**: 7:00 PM – 10:30 PM (EDT) - **Location**: Steam Whistle Brewing [...] - **Description**: This event occurred on the evening of July 11, immediately following the Best Paper Awards ceremony (which took place earlier that day at 16:00–17:30). [...] 2. **No Explicit Social Events on July 12, 2023**: - The evening of July 12, 2023, is not listed as having any official social events in the provided sources. [...] **Note**: The primary social event referenced in the sources is on July 11, not July 12. The Best Paper Awards ceremony was held on July 11, so the subsequent evening event likely refers to the Social Event on that same day. For July 12, only daytime sessions are documented. [...] `<|end_search_result|>`

Hmm, interesting. The awards were on July 11, and the search results show a social event on the evening of July 11, immediately following the awards ceremony (7:00 PM - 10:30 PM). However, the results explicitly state there were no official social events scheduled for the evening of July 12, which is technically the evening *after* the July 11 ceremony.

Given the user's phrasing "evening after", but the lack of any scheduled event on July 12 evening, it's highly probable they meant the social event that occurred on the *same evening* as the awards ceremony, which did happen *after* the ceremony concluded earlier that day. The only documented social event connected to the awards ceremony fits this description.

Therefore, the specific schedule requested corresponds to the event on July 11.

---

> The social event on the evening after the ACL 2023 best paper awards ceremony (held on July 11, 2023) was scheduled from 7:00 PM to 10:30 PM EDT at Steam Whistle Brewing / Canada's Premium Beer.

---

Table 6: A **problem-solving** example from **main reasoning process** of WebThinker on Humanity's Last Exam dataset, with special symbols used in the search queries and search results highlighted in blue box . [...] indicates omitted content.

---

**Example from Humanity's Last Exam Dataset**

---

**Question:**
Let $\Sigma_g$ denote the oriented closed surface of genus $g$. Compute the simplicial volume of $\Sigma_{31} \times \Sigma_{17}$.
**Labeled Answer:**
11520

---

**Model Output by WebThinker**

---

Okay, so I need to compute the simplicial volume of $\Sigma_{31} \times \Sigma_{17}$, where $\Sigma_g$ is an oriented closed surface of genus g.
First, I should recall the simplicial volume of a single surface $\Sigma_g$. For $g \geq 2$, the simplicial volume is $||\Sigma_g|| = 4(g-1)$. Let me quickly verify this.
<|begin_search_query|> simplicial volume of a closed surface of genus g <|end_search_query|>
<|begin_search_result|> The simplicial volume of a closed surface of genus $g \geq 2$ is indeed $4(g-1)$. This result is derived from the relationship between simplicial volume and the Euler characteristic ($\chi$) of the surface. Specifically, for a surface $\Sigma_g$ of genus $g$, the Euler characteristic is $\chi(\Sigma_g) = 2 - 2g$. The simplicial volume is given by $||\Sigma_g|| = 2|\chi(\Sigma_g)|$, leading to:

$$||\Sigma_g|| = 2|2 - 2g| = 4(g-1).$$

[...] For the product of two closed surfaces of genus $g, h \geq 2$, the simplicial volume $||\Sigma_g \times \Sigma_h||$ has known bounds related to the product of individual volumes, but the exact value requires specific theorems. [...] Sources confirm $||\Sigma_g|| = 4(g-1)$. [...] <|end_search_result|>
Okay, the formula $||\Sigma_g|| = 4(g-1)$ for $g \geq 2$ is confirmed.
Now, the core problem is the simplicial volume of the product $\Sigma_{31} \times \Sigma_{17}$. The first search mentioned bounds but not an exact formula. I need the specific rule for products of surfaces. Let me search for that.
<|begin_search_query|> simplicial volume of product of surfaces <|end_search_query|>
<|begin_search_result|> The simplicial volume of the product of two oriented closed surfaces of genus $g$ and $h$ (each $\geq 2$) is given by the formula:

$$||\Sigma_g \times \Sigma_h|| = \frac{3}{2} \cdot ||\Sigma_g|| \cdot ||\Sigma_h||.$$

Substituting the known simplicial volumes of individual surfaces ($||\Sigma_g|| = 4(g-1)$ and $||\Sigma_h|| = 4(h-1)$), this becomes:

$$||\Sigma_g \times \Sigma_h|| = \frac{3}{2} \cdot 4(g-1) \cdot 4(h-1) = 24(g-1)(h-1).$$

This result is derived from Michelle Bucher's work, leveraging the proportionality principle for manifolds covered by $\mathbb{H}^2 \times \mathbb{H}^2$. [...] <|end_search_result|>
Ah, perfect. The formula is $||\Sigma_g \times \Sigma_h|| = \frac{3}{2} \cdot ||\Sigma_g|| \cdot ||\Sigma_h|| = 24(g-1)(h-1)$.
Now I can substitute $g = 31$ and $h = 17$:
$||\Sigma_{31} \times \Sigma_{17}|| = 24 \cdot (31-1) \cdot (17-1)$
$||\Sigma_{31} \times \Sigma_{17}|| = 24 \cdot 30 \cdot 16$
Calculating this:
$24 \times 30 = 720$
$720 \times 16 = 11520$
So the simplicial volume is 11520.
The simplicial volume of an oriented closed surface $\Sigma_g$ of genus $g \geq 2$ is $4(g-1)$. For the product of two such surfaces $\Sigma_g \times \Sigma_h$, the simplicial volume is given by $\frac{3}{2}$ times the product of their individual simplicial volumes, which simplifies to $24(g-1)(h-1)$. Applying the formula for $g = 31$ and $h = 17$:

$$||\Sigma_{31} \times \Sigma_{17}|| = 24(31-1)(17-1) = 24 \cdot 30 \cdot 16 = 720 \cdot 16 = 11520.$$

Thus, the simplicial volume of $\Sigma_{31} \times \Sigma_{17}$ is 11520 .

Table 7: An example from the **Deep Web Explorer** of WebThinker on finding deadlines for the ASH 66th annual meeting, with special symbols used in the search queries, search results highlighted in `blue box`, and click actions/results highlighted in `red box`. [...] indicates omitted content. Due to space limitations, the input "Searched Web Pages" in the table are summarized.

---

**Example from Deep Web Explorer**

---

**Inputs to Deep Web Explorer:**
**- Current Search Query:**
ASH 66th annual meeting deadlines ancillary meetings late-breaking abstracts
**- Detailed Search Intent:**
The current search intent is to find the specific deadlines for requesting ancillary meetings and submitting late-breaking abstracts for the 66th ASH Annual Meeting. The search query aims to gather this information from the official ASH website or other reliable sources to ensure accuracy and relevance.
**- Searched Web Pages:**
Provided 10 web page snippets from sources like Hematology.org and Confex.com. Pages 1, 3, 5, and 7 (e.g., "Call for Late Breaking Abstracts - Hematology.org", "Call for Abstracts - confex.com") consistently stated the late-breaking abstract submission period opens October 16, 2024, and closes October 28, 2024, 11:59 p.m. Pacific Time. Page 8 ("ASH Annual Meeting Abstracts - Hematology.org") contained conflicting dates mentioning 2025, which were disregarded as likely incorrect for the 66th meeting. None of the provided snippets explicitly mentioned the deadline for ancillary meeting requests. [...]

---

**Deep Web Explorer Processing Output**

---

Okay, I need to find deadlines for ancillary meetings and late-breaking abstracts for the 66th ASH Annual Meeting.
From the initial web pages, the late-breaking abstract deadline is confirmed as October 16-28, 2024 (11:59 PM PT) from pages 1, 3, 5, and 7.
However, none of the initial pages provide the deadline for ancillary meetings. I need to search specifically for that.
`<|begin_search_query|>` ASH 66th annual meeting ancillary meetings proposal deadline `<|end_search_query|>`
`<|begin_search_result|>` Search results included a PDF link (Web Page 1) indicating the ancillary meeting deadline is **November 22, 2024, 11:59 p.m. PST**. Another page (Web Page 4) stated the request form becomes available **July 24, 2024**. Other pages mentioned meeting dates or referred to the 2025 meeting. [...] `<|end_search_result|>`
The new search results provide the ancillary meeting deadline (Nov 22, 2024) and the date the request form opens (July 24, 2024). Web Page 1 looks like the official guidelines PDF. Let me click that to confirm details.
`<|begin_click_link|>` https://www.hematology.org/-/media/hematology/files/annual-meeting/2024_ash_ancillary_meeting_guidelines_final.pdf `<|end_click_link|>`
`<|begin_click_result|>` The clicked PDF (Ancillary Meeting Guidelines for the 66th ASH Annual Meeting) confirmed: - Request Form Availability: **Wednesday, July 24, 2024, 11:00 a.m. EDT**. - Submission Deadline: **Friday, November 22, 2024, 11:59 p.m. PST**. - A non-refundable processing fee applies, tiered by event type (e.g., $200 for Member/Non-Profit, $250 for Advisory Board, $500 for Investor Meeting). - Space is limited, first-come, first-served, using designated hotels (no convention center space). - Eligibility: Organizer must be affiliated with ASH (exhibitor, sponsor, member, etc.). - Contact for questions: asheventrequests@spargoinc.com. [...] `<|end_click_result|>`
Okay, I now have confirmed information for both late-breaking abstracts (from initial search) and ancillary meetings (from the additional search and click). I can now consolidate this information.
## Final Information
After reviewing the official ASH resources and related search results, the critical deadlines for the **66th ASH Annual Meeting** are summarized below:
### **Late-Breaking Abstracts Deadlines** - **Open Date**: October 16, 2024 - **Close Date**: October 28, 2024, at **11:59 p.m. Pacific Time** - **Details**: - Requires justification for late submission. - Submission fee: **$125 USD** (non-refundable). - Max **six abstracts** accepted for presentation on **Dec 10, 2024**.
### **Ancillary Meetings Deadlines** - **Form Available**: July 24, 2024, at **11:00 a.m. EDT** - **Submission Due**: November 22, 2024, at **11:59 p.m. PST** - **Requirements**: - Organizer must be ASH-affiliated. - Tiered fees apply ($200/$250/$500 based on type). - Use approved hotels (no convention center space), first-come, first-served. - Must not conflict with official ASH programming.

Table 8: An example from the **Deep Web Explorer** of WebThinker on case studies of CLTS and Aedes mosquito control, with special symbols used in the search queries, search results highlighted in `blue box`, and click actions/results highlighted in `red box`. [...] indicates omitted content. Due to space limitations, the input "Searched Web Pages" in the table are summarized.

| **Example from Deep Web Explorer** |
| --- |

**Inputs to Deep Web Explorer:**

**- Current Search Query:**

case studies of CLTS and Aedes mosquito control in rural Southeast Asia

**- Detailed Search Intent:**

Find case studies integrating Community-Led Total Sanitation (CLTS) with Aedes mosquito control in rural Southeast Asia. The goal is to identify real-world implementations combining these programs to address waste management and water storage challenges [...] related to dengue fever.

**- Searched Web Pages:**

Provided 10 web page snippets from sources like PubMed, NCBI, Parasites & Vectors, ScienceDirect and UNAIR Repository. Several pages (1-4) focused on Aedes mosquito control methods including insecticide resistance and sterile insect techniques. Web Page 6 mentioned "government-community partnerships" for vector control in Southeast Asia. Web Page 10 discussed Aedes breeding in water containers in rural areas, though in Cameroon rather than Southeast Asia. None of the snippet information explicitly connected CLTS to Aedes control. [...]

| **Deep Web Explorer Processing Output** |
| --- |

Okay, I need to find case studies connecting CLTS and Aedes mosquito control in Southeast Asia.

From the initial web pages, I'm not seeing explicit mentions of Community-Led Total Sanitation (CLTS) being integrated with Aedes mosquito control measures. Web Page 6 mentions government-community partnerships for vector control, which is related to community engagement but not specifically CLTS. Let me check the UNAIR repository article for more details.

`<|begin_click_link|>` https://repository.unair.ac.id/128927/1/Artikel-Effectiveness.pdf `<|end_click_link|>`

`<|begin_click_result|>` The clicked PDF (from UNAIR Repository) detailed a study integrating Community-Led Total Sanitation (CLTS) with Aedes mosquito control in rural Southeast Asia (Indonesia, Vietnam, Philippines). The study combined CLTS implementation (community mobilization, latrine construction) with vector control (larvicide distribution, education). Results showed a 40% lower prevalence of Aedes breeding sites in villages with the integrated approach compared to controls, demonstrating improved sanitation and reduced mosquito breeding through community engagement. [...] `<|end_click_result|>`

The UNAIR article provides the exact case study information we're looking for, explicitly documenting CLTS integration with Aedes mosquito control in rural Southeast Asia.

I should also search for additional case studies to provide a more comprehensive answer.

`<|begin_search_query|>` CLTS and dengue prevention in rural Indonesia `<|end_search_query|>`

`<|begin_search_result|>` No additional clear case studies of CLTS explicitly integrated with Aedes control were found in the search results. Most sources discuss either CLTS for sanitation improvements or Aedes control measures separately, but not their intentional integration. The previously found UNAIR Repository article remains the most relevant source documenting this specific integrated approach in Southeast Asia. [...] `<|end_search_result|>`

The additional search did not yield further explicit case studies beyond the UNAIR Repository article.

## Final Information

Based on the analysis of web pages and further exploration, one key case study of CLTS integration with Aedes mosquito control in rural Southeast Asia was identified:

**Case Study from UNAIR Repository:** - **Location:** Rural villages in Indonesia, Vietnam, and the Philippines - **Intervention:** CLTS was combined with Aedes control measures including: - Community mobilization and latrine construction - Larvicide distribution for water container treatment - Education on identifying and eliminating breeding sites - Proper waste disposal and water storage practices - **Outcomes:** - 40% reduction in Aedes breeding sites in CLTS-integrated villages - Significant improvements in sanitation facilities - Enhanced community engagement and sustained behavior change

While Web Page 6 mentioned government-community partnerships in vector control, and Web Page 10 highlighted breeding site management (though in Cameroon), neither explicitly discussed CLTS integration with Aedes control.

The findings suggest that CLTS can be effectively combined with mosquito control by addressing the environmental conditions that enable mosquito breeding, particularly through improved waste management and water storage practices that reduce standing water sources.

Table 9: A **report generation** example from the **main reasoning process** of WebThinker on the Glaive dataset, concerning the optimization of 3D printed lattice structures for robotics. Special symbols highlight search queries/results (blue box) and writing/checking/editing actions (green box). [...] indicates omitted content.

---

**Research Question**

How can I optimize the lattice structure of a 3D printed robotic node for maximum strength-to-weight ratio while minimizing the amount of support material required, considering the limitations of FDM printing and the need for integrated connectors for inter-robot communication and power transfer in a swarm robotics application?

---

**Research Plan**

(1) Search for "FDM 3D printing limitations" to understand the constraints of the printing process.
(2) Look up "lattice structure optimization for 3D printing" to find general strategies and software tools.
[...]
(7) Look into "material properties for 3D printing lattice structures" for best materials.
(8) Find case studies on "3D printed robotic components" for real-world examples.

---

**Model Output by WebThinker**

Okay, so I need to help write a scientific article about optimizing a 3D printed robotic node's lattice structure [...] Let me start by following the research plan step by step.

`<|begin_search_query|>` FDM 3D printing limitations `<|end_search_query|>`

`<|begin_search_result|>` The limitations of FDM (Fused Deposition Modeling) 3D printing technology relevant to optimizing the lattice structure of a robotic node include:
1. **Print Orientation**: - FDM prints layer-by-layer, so orientation affects structural integrity and support requirements [...]
2. **Overhang Angles**: - Overhangs exceeding shallow angles (e.g., 45°) risk sagging [...] `<|end_search_result|>`

[... multiple additional searches performed for each research plan point ...]

Now I need to start writing the sections based on the gathered info. Let me start with the Introduction section first.

`<|begin_write_section|>` Introduction

The optimization of lattice structures in 3D printed robotic nodes is crucial for achieving high strength-to-weight ratios and minimizing support material, particularly in swarm robotics applications. Fused Deposition Modeling (FDM) printing presents unique challenges such as orientation-dependent anisotropy, limited overhang angles, and material constraints, which must be addressed to ensure structural integrity [...] `<|end_write_section|>` [... continues writing each section according to research plan ...]

`<|begin_search_query|>` material properties for 3D printing lattice structures `<|end_search_query|>`

`<|begin_search_result|>` PLA offers moderate rigidity but struggles with interlayer adhesion. TPEE provides superior energy absorption (3260 kJ/m$^3$) compared to PLA (200.90 kJ/m$^3$). TPU excels in producing flexible lattice structures, with honeycomb architectures demonstrating 30% higher rigidity than gyroid structures [...] `<|end_search_result|>`

`<|begin_write_section|>` Material Selection for Enhanced Performance

Selecting the right material is pivotal for optimizing lattice structures. PLA is cost-effective but brittle, requiring careful design to prevent delamination. TPU offers flexibility and superior energy absorption, making it ideal for connectors and flexible nodes [...] `<|end_write_section|>`

`<|begin_check_article|>` # Optimizing Lattice Structures in 3D Printed Robotic Nodes for Swarm Robotics: Balancing Strength-to-Weight Ratio and Minimizing Support Material ## Introduction ## FDM 3D Printing Limitations Impacting Lattice Design ### Print Orientation Effects [...] `<|end_check_article|>`

[... notices duplicate sections ...]

`<|begin_edit_article|>` Remove duplicate "Real-World Applications and Case Studies" section and ensure each numbered section corresponds correctly to the research plan's eight points. Also, reorder sections to follow the research plan sequence precisely. `<|end_edit_article|>`

[... notices missing conclusion after editing ...]

`<|begin_write_section|>` Conclusion

In conclusion, optimizing the lattice structure of 3D printed robotic nodes [...] `<|end_write_section|>`

`<|begin_edit_article|>` Ensure all sections maintain consistent formatting and cross-reference related concepts throughout the article. Add transition sentences between major sections to improve flow. `<|end_edit_article|>`

I have finished my work.

---

Table 10: An example of **report generation** by WebThinker: A summary outlining the structure and key sections of the final report on optimizing 3D printed lattice structures for swarm robotics.

| Final Report Summary |
| --- |

**Optimizing Lattice Structures in 3D Printed Robotic Nodes for Swarm Robotics: Balancing Strength-to-Weight Ratio and Minimizing Support Material**

**Introduction**
Swarm robotics draws inspiration from social insects. The optimization of lattice structures in 3D printed robotic nodes is crucial for achieving high strength-to-weight ratios and minimizing support material, particularly in swarm robotics applications. Fused Deposition Modeling (FDM) printing presents unique challenges [...]

**FDM 3D Printing Limitations Impacting Lattice Design**
Fused Deposition Modeling (FDM) 3D printing is widely adopted but has inherent limitations affecting lattice design, including print orientation effects, overhang limitations, material property restrictions, and layer adhesion challenges [...]

**Lattice Structure Optimization Techniques**
Optimizing lattice structures involves balancing strength, weight, and support material reduction. Key approaches include multi-objective frameworks (e.g., combining **Design of Experiments (DOE)** and **Genetic Algorithms (GA)**), selecting prominent lattice patterns (e.g., Hexagon, Diamond/Octet, BCC/SC, Fluorite), using appropriate software tools, optimizing printing parameters, minimizing support material, and integrating connectors [...]

**Key Factors Influencing Strength-to-Weight Ratio**
Achieving a high strength-to-weight ratio is essential. This involves careful material selection, lattice architecture design, applying optimization techniques, performing testing and validation, and balancing competing parameters [...]

**Reducing Support Material Through Design and Process Adjustments**
Minimizing support material is essential for cost, efficiency, and quality. Strategies include strategic design modifications, optimizing printing orientation, using advanced support types, manual support placement, stress-guided topology optimization, and integrating connectors [...]

**Integrating Connectors for Inter-Robot Communication and Power Transfer**
In swarm robotics, communication and power transfer between nodes are essential. This requires integrating suitable mechanical and electrical connectors (e.g., Box Joint, Dovetail, Snap Fits), considering material selection, integration techniques, and specific design considerations [...]

**Design Considerations Specific to Swarm Robotics**
Swarm robotics requires coordination through decentralized interactions. Design must account for communication systems, energy efficiency, mobility, structural integrity, and connector integration [...]

**Material Selection for Enhanced Performance**
Selecting appropriate materials (e.g., PLA, TPU, TPEE) is crucial. Considerations include mechanical properties, suitability for lattice structures, potential for closed-cell topologies or hybrid strategies, and specific material parameters and printing techniques [...]

**Real-World Applications and Case Studies**
Practical applications demonstrate the effectiveness of these principles, including custom protective covers, modular structures using spatial lattice printing, rapid tooling replacement, and lattice-based nodes for swarm systems [...]

**Conclusion**
Optimizing lattice structures for 3D printed swarm robots requires a holistic approach addressing FDM limitations, materials, and design. Selecting appropriate materials and lattice patterns enables high strength-to-weight ratios while minimizing support. Connector integration enhances functionality. Careful consideration of parameters, orientation, and materials leads to robust, lightweight nodes for coordinated tasks, as shown in case studies [...] Future work should focus on further optimization and exploring new materials and technologies.

