# OpenReview forum: "WebThinker: Empowering Large Reasoning Models with Deep Research Capability"
_NeurIPS.cc/2025/Conference — NeurIPS 2025 poster_

### Official Review · Reviewer_qgwT · 2025-06-30

**Clarity:** 2
**Significance:** 3
**Originality:** 2
**Rating:** 5
**Confidence:** 3

**Summary:**

The authors propose the WebThinker framework for answering user queries as well as writing reports. The framework augments an RLM with additional tools: web search capabilities to gather additional knowledge for a given query as well as writing tools for the second use case (write a specific chapter, check the report, and edit the report). The web search tool includes a search engine as well as a navigational tool and the ability to summarize the returned content to put less burden on the context of main RLM. The writing tools are implemented as language model-based tools with another language model. The framework works by using a single long context inference pass of the RLM, where the RLM is allowed to produce special tags to invoke tools with the RLM inference being suspended if necessary, while the tool is executed. Certain tools can be run concurrently. This leads to what the authors call "Autonomous Think-Search-and-Draft strategy", where the reasoning, knowledge acquisition and report writing (for the second use case) can be interleaved. The framework requires training to improve usage of the tools for which the authors use DPO where they select the preference pairs based on (in descending order) accuray, fewer tool calls and conciseness. The authors then proceed to evaluate both use cases, especially the question answering on a variety of benchmarks. Answers are submitted to a language model for accessing correctness (comparing to ground truth) or by using the LLM-as-a-Judge approach for the reports on a variety of metrics.

**Questions:**

* Please elaborate on the costs (time/Dollar amount/token usage) for the training as well as the evaluation tasks?
  * Did you conduct any experiments/analyzed the latency of the answer generation?
* What is the token limit for the language models in the tools?
* What are the differences to other web search implementations used for example in KGoT [1] or Hugging Face Agents [2]? I think [1] was inspired by [2] if I recall correctly.
* document memory M: How does this knowledge base work?
* While the authors use certain evaluation benchmarks that have not been used during the training, how much of out of distribution can they be considered?

[1] Besta et al. "Affordable AI Assistants with Knowledge Graph of Thoughts" (2025) arXiv:2504.02670

[2] Aymeric Roucher and Sergei Petrov. "Beating GAIA with Transformers Agents." (2025) https://github.com/aymeric-roucher/GAIA

**Ethical Concerns:**

["NO or VERY MINOR ethics concerns only"]

**Final Justification:**

I appreciate the additional experiments/data shown during the rebuttal phase, that illustrate that WebThinker provides improvements in quality while maintaining reasonable resource usage. The additional clarifications on the novelty convinced me, so most of my concerns regarding the manuscript have been addressed, which is why I will raise my score to "accept". I suggest however that authors working on improving the clarity of their writing before submitting the camera-ready version.

**Limitations:**

yes

**Paper Formatting Concerns:**

* Figure 3: typos - "edit**t**ing"/"writ**t**ing"
* references:
  * arXiv is cited inconsistently: sometimes "CoRR, abs/" and other times "arXiv preprint arXiv:"
  * [9], [27], [38], [40], [51], [55], [66] are missing their place of publication
  * [40]: author field is partially broken: "Qwen, :,"
  * NeurIPS is cited inconsistently: [24] vs. [41] (also [56] and [57])
  * [49] - missing URL?
  * [64] - was also published at ICLR'23
* line 725 to 727: "Here, {system_a} to {system_e} are sequentially the reports generated by the systems being evaluated." - sentence seems to broken
* page 24: Is the duplication of the initial prompt part before the json part intentional?
* checklist, question 16 (LLM usage): authors use LLMs in their writing tools as well as RLMs as coordinators and for reasoning

**Quality:**

2

**Strengths And Weaknesses:**

strengths:
* good evaluation

weaknesses:
* at times, the information regarding the workflow seems to be repeated multiple times without much additional details
* very few technical details in general (in the main body)
* limited novelty (other web search tools for language models exist, see references in questions)

page 25, lines 743/744: potential breach of anonymity - https://github.com/sunnynexus/WebThinker

---

> ### Author Rebuttal · Authors · 2025-07-31
>
> Thank you very much for your valuable and detailed review. We sincerely appreciate your constructive feedback and the opportunity to further clarify and strengthen our work. Please find our point-by-point responses below.
>
> > ### Q1: Please provide more details about the cost of training and evaluation tasks (time/dollar amount/token usage)?
>
> We provide a breakdown of the estimated costs for training, inference, and evaluation, using H100 80G GPU hours as the unit. The actual dollar amount will depend on current market prices.
>
> - **Training Cost**: This consists of two main components: data construction and model training.
>   - (1) *Data Construction*: This is the most time-consuming phase. We utilize 4 nodes, each with 8 NVIDIA H100 80G GPUs, running vLLM to serve all required models. Data collection is performed over 3 rounds, corresponding to 3 DPO training iterations. Each round of data collection takes approximately 25 hours, totaling about 2400 H100 GPU hours.
>   - (2) *Model Training*: For each DPO iteration, we use 1 node with 8 NVIDIA H100 80G GPUs. Each iteration takes about 18 hours, and with 3 rounds, this sums to roughly 432 H100 GPU hours.
>   - (3) *Total*: The overall training cost is therefore about 2832 H100 GPU hours (equivalent to 3.7 days using 4 nodes of 8 GPUs).
>
> - **Inference Cost**:
>   - Inference is performed on 1 node with 8 NVIDIA H100 80G GPUs using vLLM. For GPQA, GAIA, and WebWalkerQA, each dataset requires about 1.2–2 hours; HLE takes about 5 hours; and Glaive report generation takes about 1 hour. The total inference time is approximately 10 hours.
>
> - **Evaluation Cost**:
>   - For evaluation, we use DeepSeek-R1-671B and GPT-4o. GPT-4o processes about 1.6 million tokens, while DeepSeek-R1-671B processes about 1.8 million tokens.
>
> > ### Q2: Did you conduct any experiments or analyses regarding the latency of answer generation?
>
> Thank you for this important question. Computational overhead and efficiency are common challenges for all current deep research methods.
>
> WebThinker is designed to balance performance improvements with efficiency, which is particularly challenging when leveraging large-scale web information. Specifically, we introduced a deep web explorer module to intelligently process and integrate external web content, returning only refined, relevant information rather than indiscriminately inserting all retrieved pages into the reasoning chain. This avoids excessively long contexts and maintains reasoning coherence and efficiency. A similar approach is used for report generation, where auxiliary LLMs are employed for writing, checking, and editing, rather than inserting the entire report into the main reasoning chain.
>
> To empirically assess efficiency, we compared the inference time of different methods. All methods use 32B models as backbone and are evaluated on the GAIA benchmark with a single node of 8 NVIDIA H800 80G GPUs. The results are as follows:
>
> | Method            | Avg. Inference Time (s) | Avg. Retrievals | Avg. Seq. Length (tokens)      | Accuracy (%) |
> |-------------------|------------------------|-----------------|-------------------------------|--------------|
> | Direct Generation | 11.1                   | 0               | 5531                          | 22.3         |
> | Standard RAG      | 23.2                   | 1.0             | 9246                          | 32.0         |
> | Iterative RAG     | 34.4                   | 2.3             | 16734                         | 35.0         |
> | WebThinker        | 40.8                   | 4.9             | 8732 (main) + 17374 (aux)     | 48.5         |
>
> These results show that WebThinker achieves significantly higher answer accuracy (48.5%) while keeping inference time and sequence length within the same order of magnitude as traditional RAG methods. Notably, the main reasoning chain in WebThinker is even shorter than in Standard and Iterative RAG, which directly stack retrieved content. In practical deployment, smaller auxiliary LLMs can be used to further reduce inference time and computational cost, as these models only handle simple sub-tasks.
>
> > ### Q3: What is the token limit for the language models in the tools?
>
> We use the Qwen2.5 architecture (QwQ and Qwen2.5), which supports a maximum context length of 128k tokens.
>
> However, in practice, the actual sequence lengths are much shorter than this limit. For specific average sequence lengths, please refer to the statistical table in the answer to Q2.
>
> > ### Q4: What are the differences to other web search implementations used for example in KGoT [1] or Hugging Face Agents [2]? I think [1] was inspired by [2] if I recall correctly.
>
> Both KGoT and Hugging Face Agents are primarily based on the Surfer Agent from Hugging Face Agents for web search. The Surfer Agent provides core functionalities such as:
> - Google search (via SerpAPI)
> - Wikipedia page access
> - Web navigation (including page scrolling, keyword search, URL access, etc.)
>
> These systems employ an iterative tool-calling framework with a variety of tools, but integrating them coherently with the reasoning chain of the main model can be challenging.
>
> In contrast, WebThinker is designed to tightly couple web exploration with the reasoning process, enabling model-driven, autonomous deep exploration. It can thoroughly analyze webpage content, identify useful information, and decide whether to continue searching or follow links for deeper information. WebThinker departs from predefined workflows and emphasizes autonomy. Nevertheless, the web exploration tools in KGoT and Hugging Face Agents are valuable references, and we plan to further adapt and cite these related works.
>
> > ### Q5: Document memory M: How does this knowledge base work?
>
> In WebThinker, **Document Memory M** is a dynamic cumulative knowledge base used to store and manage all the web page information obtained through the Deep Web Explorer tool. Its working mechanism is as follows:
>
> **1. Storage Method**
> - **Source**: All web page content obtained through Deep Web Explorer search or link clicks (such as title, summary, full text summary, etc.) will be automatically stored in M.
> - **Update Timing**: After each call to the search tool ($T_{exp}$), new web page information will be appended to M, forming an incremental knowledge base.
>
> **2. Retrieval and Use**
> - **Retrieval Trigger**: When the main model (Main LRM) calls the report writing tool (such as $T_{draft}$, $T_{edit}$), the assistant model (Assistant LLM) will retrieve **Top-K relevant documents** (based on semantic similarity) from M.
> - **Retrieval Basis**: When retrieving, reference two parts:
>   - **Current Writing Instruction** (such as chapter topic).
>   - **Historical Report Content** (to avoid repetition or contradictions).
>
> **3. Role and Advantage**
> - **Global Consistency**: Ensure that information sources referenced in different chapters are unified, avoiding contradictions.
> - **Dynamic Expansion**: With the deepening of search, the knowledge base continues to enrich, supporting deep writing in subsequent chapters.
> - **Efficient Access**: Through Top-K retrieval, avoid injecting all webpage content into the main model context, reducing redundancy.
>
> Through this design, **Document Memory M** acts as a bridge between dynamic information acquisition and high-quality report generation, balancing information coverage with computational efficiency.
>
> > ### Q6: Although the authors use some evaluation benchmarks that were not used during training, to what extent can these be considered out-of-distribution?
>
> To better illustrate the contribution of each data component, we trained WebThinker on each individual dataset and evaluated performance across all benchmarks. The results are as follows:
>
> | Training Data Set | GPQA | GAIA | WebWalkerQA | HLE | Average |
> |-------------------|------|------|-------------|------|---------|
> | SuperGPQA         | 68.7 | 41.7 | 40.0        | 14.2 | 41.2    |
> | WebWalkerQA       | 65.7 | 44.7 | 45.0        | 13.0 | 42.1    |
> | OpenThoughts      | 64.1 | 43.7 | 41.5        | 13.6 | 40.7    |
> | NaturalReasoning  | 66.2 | 41.7 | 40.0        | 12.4 | 40.1    |
> | NuminaMath        | 63.6 | 39.8 | 39.0        | 13.8 | 39.1    |
> | All Data          | 71.2 | 48.5 | 46.5        | 15.8 | 45.5    |
>
> The experimental results indicate:
>
> 1. Models trained on a single dataset perform best on their corresponding evaluation task. For example, training on WebWalkerQA yields 44.7% accuracy on the WebWalkerQA evaluation set, outperforming other single-dataset trainings on that benchmark.
>
> 2. Different types of training data contribute variably to generalization. For instance, models trained on SuperGPQA and WebWalkerQA also perform well on other tasks, suggesting these datasets contain more generalizable reasoning patterns.
>
> 3. Training on the combined dataset yields the best overall performance across all evaluation tasks, with average accuracy improvements of 3–6 percentage points. This demonstrates the complementary nature of different data types in enhancing overall model performance.
>
> We hope these responses have addressed your concerns and clarified the design and evaluation of WebThinker. Thank you once again for your insightful feedback and suggestions. We look forward to further discussion.
>
> Best regards,
>
> The Authors of WebThinker

---

> > ### Author Response · Authors · 2025-08-04
> >
> > Thank you very much for your valuable feedback and time. We greatly appreciate your constructive comments and the opportunity to further clarify and strengthen our work.
> >
> > As previously discussed, utilizing a smaller auxiliary model has the potential to improve inference efficiency. Here, we present experimental results to validate this hypothesis and analyze its impact on overall system performance. We also provide a detailed efficiency evaluation for the report generation task.
> >
> > ### Efficiency in Problem-Solving Tasks
> >
> > The table below compares the efficiency and effectiveness of WebThinker on the GAIA dataset when using either a large (32B) or small (7B) auxiliary model (with QwQ-32B as the main model in all cases):
> >
> > | Method | Avg. Inference Time (s) | Avg. Retrievals | Avg. Seq. Length (tokens) | Accuracy (%) |
> > |--------|------------------------|-----------------|---------------------------|--------------|
> > | Direct Gen | 11.1 | 0 | 5531 | 22.3 |
> > | Standard RAG | 23.2 | 1.0 | 9246 | 32.0 |
> > | Iterative RAG | 34.4 | 2.3 | 16734 | 35.0 |
> > | WebThinker (32B Auxiliary Model) | 40.8 | 4.9 | 8732 (main)+17374 (aux) | 48.5 |
> > | **WebThinker (7B Auxiliary Model)** | **29.2** | **4.5** | **8596 (main)+13492 (aux)** | **46.5** |
> >
> > As shown, switching to the smaller 7B auxiliary model reduces WebThinker’s average inference time from 40.8 seconds to 29.2 seconds, representing a significant improvement in efficiency. Meanwhile, the accuracy only drops slightly (from 48.5% to 46.5%) and remains well above that of other baseline methods. This demonstrates that WebThinker can effectively balance efficiency and performance even with a smaller auxiliary model, making it suitable for scenarios with stricter requirements on inference latency and computational resources.
> >
> > ### Efficiency in Report Generation Tasks
> >
> > We further evaluated the efficiency and effectiveness of various methods on the report generation task using the Glaive dataset. All experiments were conducted on 8 NVIDIA H800 80G GPUs, with QwQ-32B as the main model. For auxiliary model, we used Qwen2.5-7B/32B for WebThinker. Overall quality (the average of comprehensiveness, thoroughness, factuality, and coherence) was used as the evaluation metric. The results are as follows:
> >
> > | Method | Avg. Inference Time (s) | Avg. Retrievals | Avg. Seq. Length (tokens) | Overall Quality (0-10) |
> > |--------|------------------------|-----------------|---------------------------|------------------------|
> > | Direct Gen | 19.8 | 0 | 6841 | 5.7 |
> > | Standard RAG | 38.5 | 1.0 | 10254 | 6.3 |
> > | Iterative RAG | 86.7 | 6.3 | 38706 | 6.9 |
> > | WebThinker (32B Auxiliary Model) | **69.2** | 8.7 | 10802 (main)+31202 (aux) | **8.1** |
> > | WebThinker (7B Auxiliary Model) | **57.8** | 8.2 | 11203 (main)+29607 (aux) | **7.7** |
> >
> > As illustrated above, WebThinker achieves superior overall quality in report generation while maintaining lower latency compared to Iterative RAG. A more detailed analysis is as follows:
> >
> > - **Latency**: With the 7B auxiliary model, WebThinker achieves an average inference time of 57.8 seconds, which is significantly lower than Iterative RAG’s 86.7 seconds. The average number of retrievals is 8.2, higher than Iterative RAG’s 6.3, indicating more extensive utilization of external information. While Direct Gen and Standard RAG have shorter inference times (19.8s and 38.5s, respectively), their low retrieval counts make them less suitable for complex report generation tasks.
> > - **Overall Quality**: WebThinker attains overall quality scores of 8.1 and 7.7, substantially outperforming Iterative RAG (6.9), Standard RAG (6.3), and Direct Gen (5.7). This demonstrates WebThinker’s ability to generate higher-quality reports and better integrate multi-round retrieval information.
> > - **Sequence Length**: For WebThinker with the 7B auxiliary model, the main reasoning chain and auxiliary content comprise 11,203 and 29,607 tokens, respectively. Although the total sequence length is considerable, the use of tool-based writing and editing effectively mitigates context redundancy and information overload in the main reasoning chain, thereby helping to reduce latency.
> >
> > These results provide further experimental evidence and analysis regarding the use of a smaller auxiliary model and the report generation task. We hope this clarifies the rationale behind WebThinker’s design choices.

---

> ### Author Response · Authors · 2025-08-04
>
> In addition to our responses to your *Questions*, we would also like to address your comments regarding *Weaknesses* and *Paper Formatting Concerns*.
>
> > Leak of Details on paper main body
>
> Thank you very much for your careful reading and feedback. Due to space limitations in the current version, we have aimed to provide more detailed information about WebThinker through the appendix and code. We will do our best to supplement more details about WebThinker and make our descriptions clearer in the updated version.
>
> > Limited Novelty
>
> We would like to further clarify the main contributions of this paper as follows:
>
> 1. We introduce WebThinker, a deep research agent that **autonomously search, deeply explore web pages, and draft research reports**, all within its thinking process. Unlike traditional predefined workflow, WebThinker enables the LRM itself to **perform actions on its own** while thinking, achieving **end-to-end task execution** in a single generation.
>
> 2. We propose a **Deep Web Explorer** that empowers LRMs with web search and navigation capabilities to deeply gather, traverse, and extract high-quality information from the web.
>
> 3. We introduce an **Autonomous Think-Search-and-Draft** strategy that enables real-time report writing during the thinking and searching process.
>
> 4. We develop **RL-based training strategies** that iteratively synthesize tool-usage preference data and apply online DPO training to enhance the LRM's tool utilization capabilities.
>
> 5. We demonstrate the effectiveness of WebThinker on **complex reasoning tasks** and **scientific report generation tasks** with both QwQ-based and DeepSeek-R1-based LRM backbones.
>
> > Link to the code repository
>
> We sincerely apologize for the oversight regarding the appendix link. Thank you for kindly pointing this out. We have tried our best to make the link invalid. We appreciate your understanding and will be more attentive to such details in the future.
>
> > Typos in Figure 3, formats in references, and sentence expression
>
> Thank you very much for your thorough review and thoughtful corrections. Your suggestions will significantly enhance the quality of our paper. In the revised version, we will ensure that all typos and reference formatting issues are properly addressed.
>
> > page 24: Is the duplication of the initial prompt part before the json part intentional?
>
> Yes, this duplication is intentional. Because multiple deep research reports generated by different models are inserted in the middle—resulting in a very long context (up to ~50k tokens)—we include the evaluation instruction at both the beginning and the end. This approach helps the model better adhere to the instruction. Although current LLMs such as DeepSeek-R1 and GPT-4o can process such long contexts, it remains best practice to reinforce the instruction in this way to ensure optimal performance.
>
> > checklist, question 16 (LLM usage): authors use LLMs in their writing tools as well as RLMs as coordinators and for reasoning
>
> Thank you for highlighting this point. We previously misunderstood the question. We will update our answer from [N/A] to [Yes] accordingly.
>
> We sincerely look forward to your continued feedback and discussion. Thank you again for your valuable time!
>
> Best regards,
>
> Authors of WebThinker

---

> ### Author Response · Authors · 2025-08-05
> **A Kind Reminder**
>
> Dear Reviewer,
>
> As the rebuttal phase is coming to an end, we would like to kindly check if our previous response has addressed your concerns. We would greatly appreciate any further comments or suggestions you may have.
>
> Thank you again for your time and consideration.
>
> Sincerely,
>
> The Authors

---

> > ### Comment · Reviewer_qgwT · 2025-08-06
> >
> > I apologize for my late reply.
> >
> > Thank you for answering my questions and for the clarifications.
> >
> > Considering the additional empirical evidence and the willingness of the authors to work with all reviewers, which will hopefully be incorporated into the final manuscript, I will increase my score.

---

> > > ### Author Response · Authors · 2025-08-06
> > >
> > > Thank you very much for your kind recognition.
> > >
> > > We truly appreciate your valuable suggestions, which have been instrumental in improving our manuscript. We will carefully incorporate all the feedback into the final version of the paper.
> > >
> > > Sincerely,
> > >
> > > The Authors

---

### Official Review · Reviewer_WwxH · 2025-07-03

**Clarity:** 3
**Significance:** 3
**Originality:** 3
**Rating:** 4
**Confidence:** 3

**Summary:**

This paper introduces WebThinker, a framework that enhances Large Reasoning Models (LRMs) with autonomous web research capabilities by integrating a Deep Web Explorer module and an Autonomous Think-Search-and-Draft strategy directly into the reasoning process. Unlike traditional RAG approaches that use predefined workflows, WebThinker allows LRMs to dynamically search the web, navigate through web pages by clicking links and buttons, and draft comprehensive research reports in real-time during their reasoning. The system operates in two modes: Problem-Solving Mode for complex reasoning tasks requiring external knowledge, and Report Generation Mode for creating detailed research documents. The authors enhance the framework through reinforcement learning using iterative online Direct Preference Optimization (DPO) to improve tool utilization. Experimental results on complex reasoning benchmarks (GPQA, GAIA, WebWalkerQA, HLE) and scientific report generation tasks demonstrate that WebThinker significantly outperforms existing methods and even some proprietary systems like o3-mini, achieving state-of-the-art performance among 32B parameter models.

**Questions:**

1. In Figure 2(b), what would happen if we replaced "Traditional LLMs" with "LRMs"? Although there are subsequent experimental validations, the figure here cannot highlight the advantages of your paper's architecture/training method because it introduces a second variable. I suggest optimizing this in future versions.

2. Line 192, why not test on AIME24/25, AMC23?

3. Line 200, for report generation, shouldn't you test the effectiveness of LLMs as evaluators, i.e., compare the difference between LLM as evaluator and human scores?

4. For particularly long reports, how should this be handled if the LLM's context window is not large enough?

5. What are the specific metrics for report evaluation? I think this should be clearly explained in the main text.

6. For the reasoning task, should we apply pass@k as a metric?

If my concerns can be addressed, I am willing to increase my score.

**Ethical Concerns:**

["NO or VERY MINOR ethics concerns only"]

**Final Justification:**

The response from the authors solves my concerns and questions. So I would like to raise my score from 3 to 4.

**Limitations:**

yes

**Quality:**

3

**Strengths And Weaknesses:**

**Strengths**

WebThinker addresses a critical limitation of current LRMs by integrating web exploration directly into the reasoning process rather than using separate preprocessing steps. The Deep Web Explorer module shows genuine innovation by enabling autonomous web navigation and link clicking, going beyond traditional RAG approaches. The experimental evaluation is comprehensive across multiple challenging benchmarks (GPQA, GAIA, WebWalkerQA, HLE) with consistent improvements over strong baselines including proprietary systems. The RL-based training strategy using iterative online DPO is well-motivated and effective. The paper achieves strong empirical results and provides clear methodology with good implementation details for reproducibility.

**Weaknesses**

The evaluation methodology has significant concerns, particularly heavy reliance on LLM-based evaluation without human baselines, which may introduce systematic biases. Figure 2(b) is misleading as it compares WebThinker (using LRMs) against traditional LLMs, introducing confounding variables. The experimental scope is limited - missing tests on established benchmarks like AIME24/25, and report generation uses only 30 questions.

---

> ### Author Rebuttal · Authors · 2025-07-31
>
> Thank you very much for your thorough review of our paper. We greatly appreciate the opportunity to address your questions and concerns. Please find our detailed responses to each of your points below.
>
> > ### Q1: In Figure 2(b), what would happen if we replaced "Traditional LLMs" with "LRMs"? Although there are subsequent experimental validations, the figure here cannot highlight the advantages of your paper's architecture/training method because it introduces a second variable. I suggest optimizing this in future versions.
>
> Thank you for your insightful question. You are correct that in Figure 2(a) and (b), the backbone model used is a traditional LLM, whereas in Figure 2(c) and (d), the backbone model is an LRM. This inconsistency may cause confusion.
>
> This is primarily because the workflow-based RAG methods depicted in Figure 2(a) and (b) are typically paired with traditional LLMs. In these approaches, each LLM generation predicts a single action for the current state based on a specific instruction (such as query planning or search necessity prediction). Each task step only requires the LLM to make a brief prediction, without the need for the LRM to generate lengthy chains of thought at every stage.
>
> In contrast, the end-to-end agentic reasoning proposed by WebThinker in Figure 2(c) is better suited for LRMs. Here, LRMs can autonomously invoke research tools throughout a single, extended reasoning process, enabling them to complete an entire task within one coherent generation.
>
> To resolve this inconsistency, we will update Figures 2(a) and (b) in the revised paper to use LRMs as the backbone model, thereby more clearly highlighting the unique advantages of WebThinker.
>
>
> > ### Q2: Why not test on AIME24/25, AMC23?
>
> The primary reason is that WebThinker is designed to enhance the LRM reasoning process by incorporating search, web exploration, and report writing capabilities, with the goal of enabling Deep Research.
>
> As demonstrated in prior work such as Search-o1 [1], experimental results indicate that mathematical tasks benefit only marginally from search operations, with relatively few search calls being made. This is because success in mathematical tasks typically depends more on the correctness of the problem-solving approach than on access to external knowledge. Consequently, we did not include mathematical tasks in our main evaluation, which is consistent with the methodology adopted by most deep research works, including OpenAI Deep Research [2], Kimi Researcher [3], Search-R1 [4], DeepResearcher [5], and WebDancer [6].
>
> To more precisely assess the impact of WebThinker’s design on reasoning tasks such as AIME24/25 and AMC23, we conducted targeted experiments, summarized as follows:
>
> | Method | AIME 2024 | AIME 2025 | AMC 2023 | Average |
> |---|--|-|-|-|
> | QwQ-32B | 76.7% (23/30) | 50.0% (15/30) | 92.5% (37/40) | 73.1% |
> | Search-o1-32B | 80.0% (24/30) | 46.7% (14/30) | 90.0% (36/40) | 74.4% |
> | WebThinker-32B | 83.3% (25/30) | 53.3% (16/30) | 92.5% (37/40) | 77.5% |
>
> From these results, we observe that on the three mathematical reasoning datasets—AIME 2024, AIME 2025, and AMC 2023—WebThinker-32B achieves performance comparable to or slightly better than QwQ-32B and Search-o1-32B, with improvements of 1–2 questions in the AIME series. This suggests that WebThinker’s design exhibits a certain degree of generalization to mathematical tasks. However, since mathematical problems generally require less external knowledge, the overall improvement remains limited.
>
> Looking ahead, we plan to integrate additional tools such as calculators and Python code executors to further enhance WebThinker’s capabilities on mathematical and other specialized tasks.
>
> > ### Q3: For report generation, should we test the effectiveness of LLMs as evaluators, i.e., compare the difference between LLM evaluators and human scores?
>
> Thank you for this important question. In our work, we employ the LLM-as-a-Judge approach, where large language models are used to evaluate reports generated by different systems in a listwise fashion, assigning comparative scores to assess their relative quality. We initially opted not to use human evaluation because many of our colleagues are already familiar with the report generation features of products like OpenAI, Gemini, and Grok Deep Research, which could introduce bias into their scoring.
>
> To validate the reliability of the LLM-as-a-Judge method, we invited four volunteers from finance, literature, and pharmacy backgrounds—individuals who rarely use Deep Research products and are thus unfamiliar with the report generation characteristics of each model—to manually score all reports (on a scale of 1–10) produced by each system. We then calculated the average score for each evaluation metric. The table below presents the results of this human evaluation:
>
> | Method | Comprehensive | Thoroughness | Factuality | Coherence | Avgerage |
> |-|-|-|-|-|-|
> | RAG-DeepSeek-R1 | 6.6 | 6.3 | 7.0 | 6.8 | 6.7 |
> | Grok3 DeeperSearch | 6.4 | 6.7 | 7.3 | 7.0 | 6.8 |
> | Gemini2.0 Deep Research | 8.0 | 7.8 | 7.6 | 7.7 | 7.8 |
> | WebThinker-32B-RL | 8.1 | 8.0 | 7.6 | 7.9 | 7.9 |
>
> As shown, the average human evaluation scores across various metrics are largely consistent with the trends observed in LLM-as-a-Judge scoring. For instance, both WebThinker-32B-RL and Gemini2.0 Deep Research significantly outperform RAG-DeepSeek-R1 and Grok3 DeeperSearch on all indicators, with WebThinker-32B-RL being comparable to or slightly surpassing Gemini2.0 in terms of comprehensiveness and thoroughness. This strongly suggests that the LLM-as-a-Judge method can accurately reflect the actual quality of the reports.
>
> One notable difference is that Grok3 DeeperSearch receives higher scores than RAG-DeepSeek-R1 in human evaluation, but lower scores in LLM-as-a-Judge assessments. This may be because Grok3 DeeperSearch produces reports that are more aligned with human reading preferences, even if the content is not as detailed as that of RAG-DeepSeek-R1.
>
> In future revisions, we will incorporate these human evaluation results into the main text, along with a more detailed discussion and analysis.
>
> > ### Q4: For particularly long reports, how should this be handled if the LLM's context window is not large enough?
>
> The report generation mechanism in WebThinker is specifically designed to address the challenge of generating entire reports in a single pass, which can easily exceed the model’s context window. In our approach, whenever the main LRM invokes the "write section" operation, the auxiliary LLM is only responsible for generating the content of the current section, rather than the full report. Similarly, for "edit section" operations, the auxiliary LLM produces only the revised content based on the relevant section of the report, not the entire document. As a result, as long as the report length remains within the LLM’s context window (for example, Qwen2.5 supports up to 128k tokens), the system functions smoothly.
>
> In practice, we have rarely encountered reports exceeding 128k tokens—our average report length is around 10k tokens. Should such a scenario arise, particularly during edit operations, we can segment the report, locate the relevant paragraph within these segments, and perform the necessary edits accordingly.
>
> > ### Q5: What are the specific metrics for report evaluation? I think this should be clearly explained in the main text.
>
> Thank you for this valuable suggestion. Previously, due to space constraints, we included all specific evaluation prompts in Appendix C.3.2 and did not elaborate on them in the main text. If possible, we will provide a detailed explanation of these metrics in the main text of the camera-ready version.
>
> > ### Q6: For the reasoning task, should we use pass@k as a metric?
>
> We currently use the Pass@1 metric, as in practical applications, user experience is typically determined by whether the system can generate a correct response in a single attempt, rather than across multiple generations. However, to better reflect the system’s potential reasoning capabilities, evaluating Pass@k for k > ### 1 is certainly meaningful.
>
> Accordingly, we also conducted experiments with Pass@3 and Pass@5, with the following results:
>
> | Method | GPQA | GAIA | WebWalkerQA | HLE | Average |
> |-|-|-|-|-|-|
> | Pass@1 | 71.2 | 48.5 | 46.5 | 15.8 | 45.5 |
> | Pass@3 | 78.3 | 56.3 | 52.8 | 19.2 | 51.7 |
> | Pass@5 | 82.8 | 59.2 | 55.9 | 21.4 | 54.8 |
>
> These results show that, across all datasets, Pass@3 and Pass@5 accuracy rates are noticeably higher than Pass@1. Specifically, Pass@3 improves by approximately 6 percentage points on average over Pass@1, and Pass@5 provides a further 3–4 percentage point increase. This demonstrates that WebThinker is capable of generating more diverse answers across multiple attempts, thereby increasing the likelihood of producing a correct response. Nevertheless, it is important to note that even with Pass@5, the improvement on challenging datasets such as HLE remains limited, highlighting the intrinsic difficulty of these tasks. This observation also points to future directions, such as end-to-end RL training and hierarchical RL training, to further enhance the model’s reasoning abilities.
>
> We hope the above responses address your questions. We sincerely appreciate your valuable feedback and continued support for WebThinker. We look forward to further feedback and discussion.
>
> Best regards,
>
> Authors of WebThinker
>
> ---
>
> ### References:
>
> [1] Search-o1: Agentic Search-Enhanced Large Reasoning Models
>
> [2] Introducing deep research.
>
> [3] Kimi-Researcher: End-to-End RL Training for Emerging Agentic Capabilities
>
> [4] Search-R1: Training LLMs to Reason and Leverage Search Engines with Reinforcement Learning
>
> [5] DeepResearcher: Scaling Deep Research via Reinforcement Learning in Real-world Environments
>
> [6] WebDancer: Towards Autonomous Information Seeking Agency

---

> > ### Comment · Reviewer_WwxH · 2025-08-03
> > **Thank you**
> >
> > Your response solves my concerns and questions. So I would like to raise my score.

---

> > > ### Author Response · Authors · 2025-08-03
> > >
> > > Thank you for your encouraging feedback and the time you dedicated to reviewing our work. We greatly appreciate your support!

---

### Official Review · Reviewer_sk4D · 2025-07-07

**Clarity:** 3
**Significance:** 3
**Originality:** 3
**Rating:** 4
**Confidence:** 5

**Summary:**

This paper introduces a novel framework named WebThinker, designed to address the limitations of Large Reasoning Models (LRMs) that stem from their reliance on static, internal knowledge when confronted with complex, knowledge-intensive tasks. WebThinker transforms an LRM into an autonomous agent capable of conducting deep research.

**Questions:**

See weaknesses above.

**Ethical Concerns:**

["NO or VERY MINOR ethics concerns only"]

**Final Justification:**

Regarding my concern about efficiency and computational cost (W1), the authors have demonstrated through new comparative experiments, particularly the one using a smaller auxiliary model, that their framework can maintain high performance while keeping latency and cost within an acceptable range.

Regarding my concern about the dependency on external tools (W2), the authors have verified through a new ablation study that their system is robust to different search engines and crawlers, with limited impact on performance.

Regarding my concern about the limitation to static web pages (W3), the authors have also provided a reasonable justification for their research scope.

Given that the authors' response and new data have successfully addressed my primary concerns, I am raising my score for this paper.

**Limitations:**

Yes

**Quality:**

3

**Strengths And Weaknesses:**

Strengths:
1. The WebThinker framework is a significant advancement over existing Retrieval-Augmented Generation (RAG) paradigms. It evolves from the conventional "retrieve-then-generate" model to a dynamic and autonomous "think-explore-generate" loop.
2. The choice of evaluation benchmarks is excellent. GPQA, GAIA, HLE, and WebWalkerQA are recognized in the community as demanding datasets for assessing advanced reasoning and information-processing capabilities.

Weaknesses:
1. WebThinker is a fairly complex framework, and the paper does not discuss the framework’s inference lantency, computational cost, or efficiency comparison with the standard RAG approach.
2. The performance of the system is highly dependent on external tools, such as the Bing search engine and the Crawl4AI crawler. The quality of the results returned by the search engine, the stability and coverage of the web crawling tools, will directly affect the final performance of WebThinker. The paper does not provide ablation experiments in this regard.
3. Webthinker is limited to processing static web pages based on text and links, which reduces its ability to be used in real-world scenarios.

---

> ### Author Rebuttal · Authors · 2025-07-31
>
> Thank you very much for your valuable and detailed review. We sincerely appreciate your constructive feedback and the opportunity to further clarify and strengthen our work. Please find our point-by-point responses below.
>
> > ### W1: WebThinker is a fairly complex framework, and the paper does not discuss the framework's inference lantency, computational cost, or efficiency comparison with the standard RAG approach.
>
> Thank you for this important question. Computational overhead and efficiency issues are common pain points for all current Deep Research related methods.
>
> WebThinker's design aims to balance performance improvement and efficiency, which is a challenging issue when leveraging large amounts of web information. Specifically, we designed an additional deep web explorer module to understand and integrate external web page information, returning refined helpful information rather than directly inserting all web pages into the reasoning chain. This would damage reasoning coherence and produce extremely long contexts, leading to poor performance and efficiency. This design philosophy is also applied to report generation, which utilizes additional LLMs for writing, checking, and editing report content instead of directly inserting the complete report into the reasoning chain.
>
> To verify this point, we conducted experiments to compare the inference time consumption of different methods. All methods use 32B models as backbone, tested on the GAIA benchmark using one node of 8 NVIDIA H800 80G GPUs. The results are as follows:
>
> | Method | Avg. Inference Time (s) | Avg. Retrievals | Avg. Seq. Length (tokens) | Accuracy (%) |
> |--------|------------------------|-----------------|---------------------------|--------------|
> | Direct Generation | 11.1 | 0 | 5531 | 22.3 |
> | Standard RAG | 23.2 | 1.0 | 9246 | 32.0 |
> | Iterative RAG | 34.4 | 2.3 | 16734 | 35.0 |
> | WebThinker | 40.8 | 4.9 | 8732 (main) + 17374 (aux) | 48.5 |
>
> From the experimental results, it can be seen that WebThinker achieves higher answer accuracy (48.5%) while maintaining inference time and sequence length at the same order of magnitude as traditional RAG methods. Notably, WebThinker's main reasoning chain length is even lower than Standard RAG and Iterative RAG methods that directly stack retrieved content. In actual deployment, we can completely adopt smaller auxiliary LLMs to further reduce inference time and computational costs, as auxiliary LLMs only handle simple sub-tasks.
>
>
> > ### W2: The performance of the system is highly dependent on external tools, such as the Bing search engine and the Crawl4AI crawler. The quality of the results returned by the search engine, the stability and coverage of the web crawling tools, will directly affect the final performance of WebThinker. The paper does not provide ablation experiments in this regard.
>
> Indeed, WebThinker's performance is affected by the type and quality of search engines, as well as the success rate of web crawling, but this is also a problem that all web search-based agents must face. Our experiments use Bing Search API and Crawl4AI crawler, and in actual deployment, we can completely use other search engines and web crawling methods.
>
> To verify the impact of search engines and crawler tools on WebThinker's performance, we conducted the following ablation experiments:
>
> We systematically evaluated WebThinker's performance on four benchmark datasets (GPQA, GAIA, WebWalkerQA, and HLE) under different search engines (Bing, Google Serper API) and whether to integrate Crawl4AI crawler tools. The experimental setup is as follows:
>
> - **Bing**: Use Bing Search API for web retrieval.
> - **Google Serper**: Use Google Serper API for web retrieval.
> - **Crawl4AI**: Based on retrieval results, further utilize Crawl4AI tools to crawl web page content, improving information coverage.
> - **Request**: Only use the requests library to obtain web page content.
>
> The table below summarizes the accuracy (Accuracy, %) of WebThinker on each dataset under different configurations:
>
> | Configuration | GPQA | GAIA | WebWalkerQA | HLE |
> |---------------|------|------|-------------|-----|
> | Bing + Crawl4AI | 70.7 | 48.5 | 46.5 | 15.8 |
> | Bing + Request | 69.2 | 47.6 | 46.8 | 15.4 |
> | Google Serper + Crawl4AI | 70.2 | 48.5 | 46.5 | 15.6 |
> | Google Serper + Request | 69.7 | 46.6 | 45.9 | 15.4 |
>
> **Analysis:**
> - **Impact of Search Engines**: Experiments show that Google Serper API and Bing Search API have similar overall performance, proving that WebThinker's performance has low dependency on search engine types and has good adaptability.
> - **Impact of Crawler Tools**: Compared to using only the Request library, after adopting Crawl4AI to crawl web page content, all datasets achieved slight improvements in accuracy. This indicates that more efficient web page acquisition capabilities do help enhance system performance, but the improvement is limited, indicating that WebThinker has relatively low dependency on specific crawler tools.
>
>
> > ### W3: WebThinker is limited to processing static web pages based on text and links, which reduces its ability to be used in real-world scenarios.
>
> WebThinker indeed primarily processes static web pages based on text and links, but this does not significantly reduce its applicability in real-world scenarios for the following reasons:
>
> First, most knowledge and information on the internet still exists in the form of text and hyperlinks. Internet content is still primarily text-based, which means WebThinker can process the vast majority of information resources on the internet.
>
> Second, our system design considers the diversity of practical application scenarios:
>
> 1. **Value of Static Content Processing**: In key application scenarios such as academic research, fact-checking, and knowledge-intensive Q&A, text information is the primary information carrier. WebThinker demonstrates excellent performance in these scenarios, as shown by the GPQA and GAIA benchmark results.
>
> 2. **Extensibility for Dynamic Web Pages**: WebThinker adopts a modular design and can extend its ability to process dynamic content through plugins. In our follow-up research, we have begun integrating JavaScript execution environments to enable the system to interact with dynamic web page elements. The current version focuses on establishing robust text understanding and link navigation foundations, which are necessary prerequisites for handling more complex web interactions.
>
> Additionally, we discuss this limitation in detail in Section 5.3 of the paper and propose clear future work directions, including integrating web rendering engines and adding multimodal understanding capabilities to expand the system's processing scope.
>
> Therefore, although the current version of WebThinker does focus on processing static web page content, this design choice is based on a trade-off between practical application needs and technical feasibility, and does not significantly limit its applicability in real-world scenarios. On the contrary, it lays the foundation for building more complex web agent interaction systems.
>
> Thank you again for your valuable comments and suggestions. We hope our responses have addressed your concerns and clarified the strengths and limitations of our work. We look forward to further feedback and discussion.
>
> Best regards,
> Authors of WebThinker

---

> > ### Author Response · Authors · 2025-08-04
> > **Additional Experimental Analysis on Efficiency**
> >
> > We sincerely appreciate your valuable feedback and the time you have devoted to our work. As previously discussed, utilizing a smaller auxiliary model has the potential to improve inference efficiency. Here, we present experimental results to validate this hypothesis and analyze its impact on overall system performance. We also provide a detailed efficiency evaluation for the report generation task.
> >
> > ### Efficiency in Problem-Solving Tasks
> >
> > The table below compares the efficiency and effectiveness of WebThinker on the GAIA dataset when using either a large (Qwen2.5-32B) or small (Qwen2.5-7B) auxiliary model (with QwQ-32B as the main model in all cases):
> >
> > | Method | Avg. Inference Time (s) | Avg. Retrievals | Avg. Seq. Length (tokens) | Accuracy (%) |
> > |--------|------------------------|-----------------|---------------------------|--------------|
> > | Direct Gen | 11.1 | 0 | 5531 | 22.3 |
> > | Standard RAG | 23.2 | 1.0 | 9246 | 32.0 |
> > | Iterative RAG | 34.4 | 2.3 | 16734 | 35.0 |
> > | WebThinker (32B Auxiliary Model) | 40.8 | 4.9 | 8732 (main)+17374 (aux) | 48.5 |
> > | **WebThinker (7B Auxiliary Model)** | **29.2** | **4.5** | **8596 (main)+13492 (aux)** | **46.5** |
> >
> > As shown, switching to the smaller 7B auxiliary model reduces WebThinker’s average inference time from 40.8 seconds to 29.2 seconds, representing a significant improvement in efficiency. Meanwhile, the accuracy only drops slightly (from 48.5% to 46.5%) and remains well above that of other baseline methods. This demonstrates that WebThinker can effectively balance efficiency and performance even with a smaller auxiliary model for web page exploration, making it suitable for scenarios with stricter requirements on inference latency and computational resources.
> >
> > ### Efficiency in Report Generation Tasks
> >
> > We further evaluated the efficiency and effectiveness of various methods on the report generation task using the Glaive dataset. All experiments were conducted on 8 NVIDIA H800 80G GPUs, with QwQ-32B as the main model. For auxiliary model, we used Qwen2.5-7B/32B for WebThinker. Overall quality (the average of comprehensiveness, thoroughness, factuality, and coherence) was used as the evaluation metric. The results are as follows:
> >
> > | Method | Avg. Inference Time (s) | Avg. Retrievals | Avg. Seq. Length (tokens) | Overall Quality (0-10) |
> > |--------|------------------------|-----------------|---------------------------|------------------------|
> > | Direct Gen | 19.8 | 0 | 6841 | 5.7 |
> > | Standard RAG | 38.5 | 1.0 | 10254 | 6.3 |
> > | Iterative RAG | 86.7 | 6.3 | 38706 | 6.9 |
> > | WebThinker (32B Auxiliary Model) | **69.2** | 8.7 | 10802 (main)+31202 (aux) | **8.1** |
> > | WebThinker (7B Auxiliary Model) | **57.8** | 8.2 | 11203 (main)+29607 (aux) | **7.7** |
> >
> > As illustrated above, WebThinker achieves superior overall quality in report generation while maintaining lower latency compared to Iterative RAG. A more detailed analysis is as follows:
> >
> > - **Latency**: With the 7B auxiliary model, WebThinker achieves an average inference time of 57.8 seconds, which is significantly lower than Iterative RAG’s 86.7 seconds. The average number of retrievals is 8.2, higher than Iterative RAG’s 6.3, indicating more extensive utilization of external information. While Direct Gen and Standard RAG have shorter inference times (19.8s and 38.5s, respectively), their low retrieval counts make them less suitable for complex report generation tasks.
> > - **Overall Quality**: WebThinker attains overall quality scores of 8.1 and 7.7, substantially outperforming Iterative RAG (6.9), Standard RAG (6.3), and Direct Gen (5.7). This demonstrates WebThinker’s ability to generate higher-quality reports and better integrate multi-round retrieval information.
> > - **Sequence Length**: For WebThinker with the 7B auxiliary model, the main reasoning chain and auxiliary content comprise 11,203 and 29,607 tokens, respectively. Although the total sequence length is considerable, the use of tool-based writing and editing effectively mitigates context redundancy and information overload in the main reasoning chain, thereby helping to reduce latency.
> >
> > These results provide further experimental evidence and analysis regarding the use of a smaller auxiliary model and the report generation task. We hope this clarifies the rationale behind WebThinker’s design choices.
> >
> > We sincerely look forward to your continued feedback and discussion. Thank you again for your valuable time!
> >
> > Best regards,
> >
> > Authors of WebThinker

---

> ### Author Response · Authors · 2025-08-05
> **A Kind Reminder**
>
> Dear Reviewer,
>
> As the rebuttal phase is coming to an end, we would like to kindly check if our previous response has addressed your concerns. We would greatly appreciate any further comments or suggestions you may have.
>
> Thank you again for your time and consideration.
>
> Sincerely,
>
> The Authors

---

### Official Review · Reviewer_6BBj · 2025-07-13

**Clarity:** 3
**Significance:** 2
**Originality:** 2
**Rating:** 4
**Confidence:** 3

**Summary:**

The work proposes WebThinker, which augments large reasoning models with autonomous web exploration and report generation functiom. The models can dynamically search the web, navigate pages, and draft or revise content during the reasoning process. The difference with RAG is that, WebThinker interleaves reasoning, searching, and writing in an integrated loop. It introduces a Deep Web Explorer and specialized drafting tools, and further leverages RL to optimize tool usage based on preference feedback. The system achieves better performance on challenging benchmarks, including GPQA and Humanity’s Last Exam, outperforming both open-source and proprietary baselines.

**Questions:**

See Weakness

**Ethical Concerns:**

["NO or VERY MINOR ethics concerns only"]

**Final Justification:**

I reduced my confidence by 1, as I am not sure about the novelty claim. The authors said "However, prior to early this year, most existing RAG methods did indeed rely on predefined workflows.", I don't think this is a good reason for one to say they are novel and the novelty exists before a specific time frame. Many industry solutions and agentic framework has already incorporated similar techniques.

**Limitations:**

The authors point out several limitations in the conclusion section:
- WebThinker cannot process multimodal information such as images and videos
- It currently supports only a limited set of tools
- Extending WebThinker to support GUI-based web exploration would enable it to handle more complex and real-world interactive tasks.

**Quality:**

3

**Strengths And Weaknesses:**

**Strengths:**
- The presentaion and flow is clear
- The experiments are pretty comprehensive

**Weaknesses**:
- **Fair Comparison**: I am not sure if the comparison is entirely fair between WebThinker and baselines. In Figure 2, there're multiple traditional LLMs decide whether more search is needed. Moreover, with multiple retrieved content blocks accumulated in the context, the cost or input size can be huge, which is also identified by authors: `Justification: Due to the high computational cost of our experiments, we did not report error bars.`. Simultaneously, the method needs to maintain a written report. Can you provide more justification into the cost in the inference time, and how does this compared to other models?
- **Clarity**: The authors say "The objective is to generate a comprehensive answer solution for a given task query q" (Line 96). What does "comprehensive" mean here specifically? Why is this important in all cases?
- **Transparency:** Following Weaknesses point 1, how redundant is the search process? Would it be possible that the agent decides to keep searching while this is actually not necesary? While I see the author penalize such redundancy with the DPO where shorter trajectories are preferred, I imagine it might still happen.
- **Training details**: It is not clear how long and how many trajectories does the model need to be effective? In Section 3.5, what are the percentages of different types of DPO pairs?
- **(Minor) Novelty**: One has to admit that many components in WebThinker are not new and explored in the previous works, but in general this looks like a useful approach. Therefore the issue is a minor concern to me.

---

> ### Author Rebuttal · Authors · 2025-07-31
>
> Thank you very much for your detailed review. We have addressed your concerns point by point and further supported our responses with comprehensive quantitative analyses:
>
> > ### Q1: Fair Comparison: Concerns about computational costs and inference time comparisons between WebThinker and baselines
>
> Thank you for raising this important point. Computational overhead and efficiency are indeed common challenges faced by all current Deep Research methods.
>
> WebThinker is designed to strike a balance between performance gains and efficiency, which is particularly challenging when incorporating large volumes of web information. To address this, we introduced a deep web explorer module that intelligently understands and integrates external web content, returning distilled and relevant information rather than indiscriminately inserting entire web pages into the reasoning chain. Directly including all retrieved content would disrupt reasoning coherence and result in excessively long contexts, ultimately degrading both performance and efficiency. This design philosophy also extends to report generation, where we employ additional LLMs for writing, checking, and editing report content, instead of embedding the entire report into the main reasoning chain.
>
> To empirically validate this approach, we conducted experiments comparing the inference time of different methods. All methods utilized 32B models as the backbone and were evaluated on the GAIA benchmark using a single node equipped with 8 NVIDIA H800 80G GPUs. The results are summarized below:
>
> | Method         | Avg. Inference Time (s) | Avg. Retrievals | Avg. Seq. Length (tokens) | Accuracy (%) |
> |----------------|------------------------|-----------------|---------------------------|--------------|
> | Direct Gen     | 11.1                   | 0               | 5531                      | 22.3         |
> | Standard RAG   | 23.2                   | 1.0             | 9246                      | 32.0         |
> | Iterative RAG  | 34.4                   | 2.3             | 16734                     | 35.0         |
> | WebThinker     | 40.8                   | 4.9             | 8732 (main)+17374 (aux)   | 48.5         |
>
> As shown, WebThinker achieves significantly higher answer accuracy (48.5%) while keeping inference time and sequence length within the same order of magnitude as traditional RAG methods. Notably, the main reasoning chain length in WebThinker is even shorter than that of Standard RAG and Iterative RAG, which directly stack retrieved content. In practical deployment, we can further reduce inference time and computational costs by employing smaller auxiliary LLMs, since these only handle relatively simple sub-tasks.
>
>
> > ### Q2: Clarity: The authors say "The objective is to generate a comprehensive answer solution for a given task query q" (Line 96). What does "comprehensive" mean here specifically? Why is this important in all cases?
>
> In this context, "comprehensive" encompasses both the entire reasoning process and the final answer or report. We believe that, whether in problem-solving or report generation scenarios, both the reasoning process and the final output are essential, which is why we use this term.
>
> However, we acknowledge that this expression could be clearer. A more precise formulation would be: ```The objective is to generate a comprehensive reasoning process and answer for a given task query q, guided by an instruction I```, which better conveys our focus on both the completeness of the reasoning process and the thoroughness of the final answer.
>
> > ### Q3: Transparency: Concerns about redundancy in the search process and potential for unnecessary searches
>
> Good question. Indeed, redundant search calls by the Search Agent can lead to unnecessary computational costs, making efficiency a critical concern in this field. To address this, WebThinker mitigates redundancy at the training data level by constructing preference pairs: reasoning trajectories with fewer and more accurate search calls are treated as positive examples, while those with incorrect or excessive search calls are treated as negative examples.
>
> To more concretely illustrate the impact of this design on search efficiency, we conducted the following ablation experiments:
>
> | Model Variant | Avg. Search Count | Effective Search Ratio (%) | WebWalkerQA Accuracy (%) |
> |---------------|------------------|---------------------------|-------------------------|
> | WebThinker (no DPO) | 7.3 | 42.5 | 41.7 |
> | WebThinker (correctness DPO only) | 6.8 | 45.3 | 45.2 |
> | WebThinker (full DPO) | 4.9 | 68.7 | 46.5 |
>
> As shown, the full DPO training strategy significantly reduces the average number of search calls (from 7.3 to 4.9, a 32.9% reduction) and substantially increases the effective search ratio (from 42.5% to 68.7%). Here, the "effective search ratio" refers to the proportion of searches along the labeled path relative to the total number of searches, serving as a measure of the model’s search efficiency. These results indicate that our training strategy effectively guides the model to search more selectively, thereby reducing redundant operations.
>
> Importantly, this improvement in search efficiency does not come at the expense of performance. On the contrary, accuracy on the WebWalkerQA benchmark increases by 3.8 percentage points. This demonstrates that our approach not only enhances efficiency but also improves the model’s overall reasoning ability, enabling it to better determine when additional information is needed and what content should be retrieved.
>
> > ### Q4: Training details: It is not clear how long and how many trajectories does the model need to be effective? In Section 3.5, what are the percentages of different types of DPO pairs?
>
> Below are the detailed quantities and proportions of each data type in our training set:
>
> | Dataset | Training Pairs | Percentage |
> |---------|---------------|------------|
> | SuperGPQA | 2,973 | 32.0% |
> | WebWalkerQA | 2,891 | 31.1% |
> | OpenThoughts | 1,637 | 17.6% |
> | NaturalReasoning | 1,108 | 11.9% |
> | NuminaMath | 842 | 9.1% |
> | **Total** | **9,451** | 100% |
>
> As shown, SuperGPQA and WebWalkerQA contribute 2,973 and 2,891 data points, respectively, while OpenThoughts, NaturalReasoning, and NuminaMath provide 1,637, 1,108, and 842 data points, for a total of 9,451. These cover a wide range of scientific, knowledge-intensive, mathematical, logical reasoning, and open-ended tasks, aiming to encompass as many real-world task types as possible.
>
> To investigate the impact of training duration and data volume on model effectiveness, we trained the model on a single node with 8 NVIDIA H800 80G GPUs using the above 9,451 data points (batch size 32), and evaluated performance on the GAIA and WebWalkerQA datasets every 200 steps. The table below shows accuracy progression with training steps:
>
> | Step | GPQA | GAIA | WebWalkerQA | HLE  |
> |------|------|------|-------------|------|
> | 0    | 68.7 | 44.7 | 41.9        | 13.0 |
> | 200  | 69.1 | 46.5 | 44.1        | 14.4 |
> | 400  | 69.1 | 47.6 | 44.3        | 15.2 |
> | 600  | 70.2 | 48.5 | 45.4        | 15.8 |
> | 800  | 71.2 | 48.5 | 46.5        | 15.8 |
>
> The results indicate that as training progresses, the model’s accuracy on both datasets steadily improves and converges around 800 steps, suggesting that the current data volume and training epochs are sufficient to fully realize the model’s potential. Notably, even after just 200 steps, the model exhibits significant performance gains, demonstrating that the current training data is already highly effective.
>
> > ### Q5: (Minor) Novelty: One has to admit that many components in WebThinker are not new and explored in the previous works, but in general this looks like a useful approach. Therefore the issue is a minor concern to me.
>
> Thank you for raising this point. The primary goal of WebThinker is to equip large reasoning models with deep research capabilities, with web exploration and report writing being the two most critical functionalities. While prior work has addressed these tasks, most existing methods rely on predefined workflows, making it difficult to seamlessly integrate with extended reasoning processes.
>
> WebThinker addresses this by abstracting web exploration, report writing, and editing as "research tools," enabling the reasoning model to autonomously invoke these tools within a single, end-to-end agentic reasoning process to accomplish complex tasks. This approach—especially in the context of report generation and tool integration—remains underexplored in the current literature.
>
> We hope the above clarifications and experimental results address your questions. Thank you again for your valuable feedback.
>
> Best regards,
> Authors of WebThinker

---

> > ### Author Response · Authors · 2025-08-04
> > **Additional Experimental Analysis on Efficiency**
> >
> > We sincerely appreciate your valuable feedback and the time you have dedicated to our work. As previously discussed, employing a smaller auxiliary model can potentially enhance inference efficiency. Here, we present experimental results to validate this hypothesis and analyze its impact on overall system performance. Additionally, we provide a detailed efficiency evaluation and analysis for the report generation task.
> >
> > ### Efficiency in Problem Solving Tasks
> >
> > The table below compares the efficiency and effectiveness of WebThinker on the GAIA dataset when using either a large (Qwen2.5-32B) or small (Qwen2.5-7B) auxiliary model (with QwQ-32B as the main model in all cases):
> >
> > | Method | Avg. Inference Time (s) | Avg. Retrievals | Avg. Seq. Length (tokens) | Accuracy (%) |
> > |--------|------------------------|-----------------|---------------------------|--------------|
> > | Direct Gen | 11.1 | 0 | 5531 | 22.3 |
> > | Standard RAG | 23.2 | 1.0 | 9246 | 32.0 |
> > | Iterative RAG | 34.4 | 2.3 | 16734 | 35.0 |
> > | WebThinker (32B Auxiliary Model) | 40.8 | 4.9 | 8732 (main)+17374 (aux) | 48.5 |
> > | **WebThinker (7B Auxiliary Model)** | **29.2** | **4.5** | **8596 (main)+13492 (aux)** | **46.5** |
> >
> > As shown, when switching to the smaller 7B auxiliary model, WebThinker’s average inference time drops from 40.8 seconds to 29.2 seconds, representing a notable improvement in efficiency. Meanwhile, the accuracy decreases only slightly (from 48.5% to 46.5%) and remains well above that of other baseline methods. This demonstrates that WebThinker can effectively balance efficiency and performance even with a smaller auxiliary model for web page exploration, making it suitable for scenarios with stricter requirements on inference latency and computational resources.
> >
> > ### Efficiency in Report Generation Tasks
> >
> > We further evaluated the efficiency and effectiveness of various methods on the report generation task using the Glaive dataset. All experiments were conducted on 8 NVIDIA H800 80G GPUs, with QwQ-32B as the main model. For auxiliary model, we used Qwen2.5-7B/32B for WebThinker. Overall quality (the average of comprehensiveness, thoroughness, factuality, and coherence) was used as the evaluation metric. The results are as follows:
> >
> > | Method | Avg. Inference Time (s) | Avg. Retrievals | Avg. Seq. Length (tokens) | Overall Quality (0-10) |
> > |--------|------------------------|-----------------|---------------------------|------------------------|
> > | Direct Gen | 19.8 | 0 | 6841 | 5.7 |
> > | Standard RAG | 38.5 | 1.0 | 10254 | 6.3 |
> > | Iterative RAG | 86.7 | 6.3 | 38706 | 6.9 |
> > | WebThinker (32B Auxiliary Model) | **69.2** | 8.7 | 10802 (main)+31202 (aux) | **8.1** |
> > | WebThinker (7B Auxiliary Model) | **57.8** | 8.2 | 11203 (main)+29607 (aux) | **7.7** |
> >
> > As illustrated above, WebThinker achieves superior overall quality in report generation while maintaining lower latency compared to Iterative RAG. A more detailed analysis is as follows:
> >
> > - **Latency**: With the 7B auxiliary model, WebThinker achieves an average inference time of 57.8 seconds, which is significantly lower than Iterative RAG’s 86.7 seconds. The average number of retrievals is 8.2, higher than Iterative RAG’s 6.3, indicating more extensive utilization of external information. While Direct Gen and Standard RAG have shorter inference times (19.8s and 38.5s, respectively), their low retrieval counts make them less suitable for complex report generation tasks.
> > - **Overall Quality**: WebThinker attains overall quality scores of 8.1 and 7.7, substantially outperforming Iterative RAG (6.9), Standard RAG (6.3), and Direct Gen (5.7). This demonstrates WebThinker’s ability to generate higher-quality reports and better integrate multi-round retrieval information.
> > - **Sequence Length**: For WebThinker with the 7B auxiliary model, the main reasoning chain and auxiliary content comprise 11,203 and 29,607 tokens, respectively. Although the total sequence length is considerable, the use of tool-based writing and editing effectively mitigates context redundancy and information overload in the main reasoning chain, thereby helping to reduce latency.
> >
> > The above results provide additional experimental evidence and analysis regarding the use of a smaller auxiliary model and the report generation task. We hope this helps clarify the design choices behind WebThinker.
> >
> > We sincerely look forward to your further feedback and discussion. Thank you again for your valuable time!
> >
> > Best regards,
> >
> > Authors of WebThinker

---

> ### Author Response · Authors · 2025-08-05
> **A Kind Reminder**
>
> Dear Reviewer,
>
> As the rebuttal phase is coming to an end, we would like to kindly check if our previous response has addressed your concerns. We would greatly appreciate any further comments or suggestions you may have.
>
> Thank you again for your time and consideration.
>
> Sincerely,
>
> The Authors

---

> ### Comment · Reviewer_6BBj · 2025-08-05
>
> Thanks the author for the response.
> While I still hold a bit of the concern on the novelty part and I think the statement "most existing methods rely on predefined workflows" is not entirely fair, I keep my original recommendation for now.

---

> > ### Author Response · Authors · 2025-08-05
> >
> > Dear Reviewer,
> >
> > Thank you for your reply!
> >
> > We’d like to clarify our point further. You're right that this statement may not fully reflect the **current** landscape, as many follow-up works have now advanced this paradigm. However, **prior to early this year**, most existing RAG methods did indeed rely on predefined workflows.
> >
> > We sincerely appreciate your time and valuable feedback.
> >
> > Best regards,
> >
> > The Authors

---

### Note · Authors · 2025-08-14

Dear area chair and all reviewers,

We sincerely appreciate the time and effort to review our paper. We have carefully addressed each point raised and conducted additional experiments during the rebuttal phase. Below, we provide a clear summary of the reviewers' main concerns, engagement, and the main contributions:

> ### Main Concerns Raised by the Reviewers
1. **Computational Efficiency and Inference Time**
- Extensive experiments on both *problem-solving* and *report generation* tasks show that WebThinker achieves significantly higher accuracy while maintaining inference time comparable to traditional RAG methods.
- Using smaller auxiliary models (7B) further reduces inference time with minimal accuracy loss.
2. **Training Data, Cost, and Model Performance**
- We provide a detailed breakdown of 9,451 training pairs from 5 datasets.
- We provide detailed cost analysis for all experiments.
- We analyze the contribution of each data component to overall performance.
3. **Evaluation Methods and Results**
- For problem-solving tasks, we provide detailed Pass@[1,3,5] results and performance on math tasks (AIME, AMC).
- For report generation, we evaluate report quality with four human volunteers from different domains and obtained similar conclusions, confirming the reliability of our LLM-as-Judge method.

> ### Reviewer's Engagement
- **Reviewer 6BBj, WwxH and qgwT** provided detailed reviews, and all replied to our rebuttal on time.
- **Reviewer sk4D** hasn't provided any feedback and we truly hope to get your feedback soon.

> ### Main Contributions of this Paper
1. We introduce **WebThinker**, a deep research agent that **autonomously search, deeply explore web pages, and draft research reports, all within its thinking process.**
2. We propose a **Deep Web Explorer** that empowers LRMs with web search and navigation capabilities to deeply gather, traverse, and extract high-quality information from the web.
3. We introduce an **Autonomous Think-Search-and-Draft strategy** that enables real-time report writing during the thinking and searching process.
4. We develop **RL-based training strategies** that iteratively synthesize tool-usage preference data and apply online DPO training to enhance the LRM's tool utilization capabilities.
5. Extensive experiments show the effectiveness of WebThinker on various **complex reasoning tasks** and **scientific report generation tasks** with both QwQ-based and DeepSeek-R1-based LRM backbones.

Sincerely,

The Authors

---

### Decision · Program_Chairs · 2025-09-17

**Decision:**

Accept (poster)

**Comment:**

This paper introduces WebThinker, a framework that augments Large Reasoning Models (LRMs) with autonomous web exploration and report generation capabilities through a Deep Web Explorer module and an Autonomous Think-Search-and-Draft strategy. WebThinker enables dynamic web search, navigation, and real-time report drafting within the reasoning process, enhanced via reinforcement learning with iterative online DPO.

The paper demonstrates strong empirical results across multiple benchmarks (GPQA, GAIA, WebWalkerQA, HLE) and report generation tasks. All four reviewers acknowledged the comprehensive evaluation and practical value of the system. During the rebuttal, authors provided extensive additional experiments addressing efficiency concerns, showing that WebThinker achieves 48.5% accuracy on GAIA with inference time comparable to traditional RAG methods (40.8s vs 34.4s for Iterative RAG), and that using smaller auxiliary models (7B) can reduce latency to 29.2s with minimal accuracy loss. Authors also validated their LLM-as-Judge evaluation through human studies with four volunteers from different domains, confirming similar quality rankings.

However, reviewers raised concerns about novelty and clarity that were partially addressed. Reviewer 6BBj noted that many components are not new, though acknowledged the system's usefulness. The claim that "most existing methods rely on predefined workflows" was questioned as not entirely fair given recent advances in the field. Reviewer qgwT pointed out limited technical details in the main body and repetitive information presentation. While authors clarified their contributions, the novelty concerns regarding the integration of existing techniques remain valid.

The consensus among the engaged reviewers supports acceptance. The paper makes solid empirical contributions with practical value, despite limited conceptual novelty. The extensive experiments, including those added during rebuttal, demonstrate the system's effectiveness and efficiency trade-offs.

The recommendation is to accept this paper.